# Symbolic Regression with Self-Supervised Heuristic Beam Search

## Abstract

*Symbolic Regression* (SR) aims to discover simple and interpretable mathematical expressions that explain observed data, making it a powerful tool for scientific discovery. In this work, we introduce a conceptually simple SR method that is both sample-efficient with respect to observed data points and self-supervised on large-scale synthetic data. By design, our approach favors parsimony, yielding interpretable and concise expressions. We focus on problems with exact solutions, evaluating our method on datasets containing physical laws and dynamical equations. Our results demonstrate that combining beam search with a learned heuristic achieves competitive performance compared to existing methods in SR-Bench. Additionally, our approach effectively handles expressions with constants, a common challenge in the SR field. Finally, we provide a comprehensive scalability analysis across four key dimensions: (i) expression length, (ii) number of variables, (iii) number of domains, and (iv) number of observed data points.

## 1 Introduction

In Machine Learning, many models are designed to achieve low training error and perform well in unseen but similar data. Yet, fitness to data is not the only important attribute. Some applications require interpretability: models must be meaningful in terms of familiar constructs. Another desirable quality is to have Out-Of-Distribution (OOD) generalization. In this context, **Symbolic Regression** (SR) is the task of finding mathematical expressions that fit the data and are as simple as possible. In Physics and other natural sciences, interpretability is commonly accompanied by OOD generalization, as laws of nature have been widely tested. This makes SR a good candidate for finding scientific insight from data. Other areas that can benefit from SR include medicine and finance (Jobin et al., 2019; Rudin, 2019), which are critical and high-stakes.

Formally, given a *domain* set of data points $\mathcal{D} := \{(\mathbf{x}_i, y_i)\}_{1 \leq i \leq n}$ consisting of paired features $\mathbf{x}_i$ and target values $y_i$, the goal of SR is to find a mathematical expression $E$ such that $E(\mathbf{x}_i) \approx y_i$ and $E$ is as simple as possible (e.g. it has a small number of symbols from a pre-defined vocabulary). In the case where an *exact* solution $F$ exists, it is required that $E \equiv F$ up to some tolerance on constant values that may appear (e.g. $1.5x \cdot x + 2.0001$ and $1.4999x^2 + 2$ may be considered equivalent).

In this paper, we present **HTSSR: HeurisTic beam Search Symbolic Regression**, a new method for SR that learns, in a self-supervised way, a precedence relation among expressions to guide a beam search algorithm. We detail key design choices that make our results possible, investigate the scalability of the search and its ability to work with only a few data points, and compare HTSSR against existing methods on SRBench (Cava et al., 2021). Besides being a new method for SR compared against existing work, the **contributions** of HTSSR have many facets:

- **A shift from the current generative approach**: Our heuristic model is solely dedicated to the task of attributing scores for the search elements, with no need to predict symbols. The expression formation happens by expanding preceding expressions with pre-defined rules, allowing great control over expression generation.

- **Independence from fitness to data**: When training the heuristic model, there is no need to use numeric fitness to data information, as it can be ambiguous and unstable. The model can be trained self-supervised with virtually infinite synthetic expressions.

- **Clean, simple, and modular design**: Our design allows easy and free customization of symbolic vocabulary, operator definition, generation rules, heuristic model, and search algorithm. Although we use beam search in this work, it can be easily replaced by other algorithms such as stochastic search or MCTS.

- **A new and elegant way to frame the search heuristic**: We demonstrate that learning the guiding heuristic is equivalent to learning a binary classification task. This contrasts with existing learning-based methods with complicated training processes or heavy reward engineering.

- **Almost no assumption about data distribution**: Because the heuristic can be dedicated to a specific dataset, it is not necessary to make assumptions about the distribution of input data and it is easy to avoid overfitting. The only exception is the distribution of constants that may appear.

- **Robustness to noise and efficiency with scarce data**: Expressions can be found even in the presence of noise or few available data points. This might be useful in real world applications where data is scarce or considerably noisy.

## 2 RELATED WORK

**Genetic Programming (GP)** was the first note-worthy way to approach SR and many SR methods fall into this category. Early works include (Koza, 1989; 1990), which deal with Program Synthesis, a superclass of SR in a sense. More recent applications of GP to SR are (Keijzer, 2003; Vladislavleva et al., 2009; Schmidt & Lipson, 2009; Korns, 2011; Uy et al., 2011; Jin et al., 2020). GP techniques are known to be easily parallelized and have high parallelism, allowing for the evaluation of a high number of expressions. One downside of GP methods is that they are not robust in cases involving hyperparameters (Petersen et al., 2021). Hybrid approaches, like those proposed in (Mundhenk et al., 2021; Kamienny et al., 2023), combine Deep Learning and GP by letting one or more learned models perform sub-tasks of the GP search, like population seeding, mutation, and selection. (Mundhenk et al., 2021) combines GP with Deep Learning by seeding the GP search with expressions from the learned model. In principle, the learned models help guide GP to more promising regions in search space. Similarly to (Petersen et al., 2021), the model is trained with Reinforcement Learning with the reward signal based on the fitness to data. A clear disadvantage is that a supervision/reward signal based on numerical fit means very different things depending on the context. For instance, the same numerical error may come from a candidate solution that is very close or very far in the space of discrete expressions. In contrast, the supervision of our heuristic model is a simple binary value indicating a precedence relation between pairs of elements, having simple optimization and using well-stabilised binary cross-entropy loss.

**The application of Deep Learning to SR** has early examples like (Kusner et al., 2017; Sahoo et al., 2018; Alaa & van der Schaar, 2019). The work (Udrescu & Tegmark, 2020) is possibly the first to show notable progress of Deep Learning in SR. It approaches Symbolic Regression mostly by simplifying a problem into subproblems. (Cranmer et al., 2020; Bendinelli et al., 2023) also allows for the inclusion of simplifying assumptions or prior knowledge. Even though problem simplification should be used in expression discovery, it needs domain-specific knowledge and even so there is always some remaining search space of possible solutions. Instead, we focus on the search guidance approach and let problem simplification for further study.

**Regarding neural architecture**, many recent works employ Transformer architecture (d'Ascoli et al., 2024; Shojaee et al., 2023b; Kamienny et al., 2023; Lalande et al., 2023; Valipour et al., 2021; Biggio et al., 2021). Even though we do use Transformer layers in the last part of our neural networks to process expressions as sequences, we do not use those as a generative model. (Petersen et al., 2021) uses RNN architecture while (Cranmer et al., 2020) uses GNN. Some works hard-code symbolic operations inside the neural networks in order to recover an expression after training (Kim et al., 2021), (Kubalík et al., 2023).

**Generative methods** can be divided in two types: (i) a generative model is trained to infer the desired expressions as sequences of tokens (Kamienny et al., 2022; Biggio et al., 2021; Petersen et al., 2021; Vastl et al., 2022; d'Ascoli et al., 2024); (ii) a decoding or search strategy is added to find expressions using token probabilities from a model of the first type (Shojaee et al., 2023a; Bendinelli et al., 2023; Hayes et al., 2025). The runtime of the methods in the first category is

usually low, although independent sampling may produce inferior solutions. The methods in the second category take advantage of post-training time and employ sophisticated search strategies, such as MCTS. Although MCTS is a popular choice, long lookaheads may yield large expressions, while a beam search might be more parsimonious. Additionally, some methods can add RL after the first training and before the search phase (Hayes et al., 2025), or have RL but no extra search strategy at all (Petersen et al., 2021). With multiple phases, a training pipeline can become complicated. RL methods may rely heavily on reward engineering with the fitness to data signal, which is ambiguous and unstable.

TPSR (Shojaee et al., 2023a) proposes using the token probabilities from some pre-trained SR model like E2ESR (Kamienny et al., 2022) in order to perform MCTS. While the source model is trained to generate correct expressions in a single shot, it needs more inference runs to achieve success in practice. TPSR brings a smarter search strategy than pure random trials or token-level beam search. The only similarity to our work is the existence of a training phase followed by a search phase. Even if the source model is trained in a self-supervised way, it is a different setup because our heuristic model does not produce any symbol itself, but rather just a binary signal. Also, one disadvantage of long lookaheads of the MCTS in TPSR is that the expressions found tend to be large. Even in the Feynman datasets with most ground-truth solutions having fewer than 15 symbols, TPSR finds non-exact solutions with more than 50 symbols.

A key difference between the generative models and our heuristic model is that we do not have an explicit distribution over tokens. Instead, the output of the model is a score that can be used to prioritize elements in a search. Also, the expression generation in our method is independent of any parametric model: it happens by applying pre-defined grammar-like generation rules and is very fast by means of its simplicity. Another advantage of our method lies in its strong theoretical justification. Like (Yu et al., 2025), our search objective contains an optimal substructure (Cormen et al., 2009): if the heuristic prediction is the true precedence relation (Section 3), a beam search that prioritizes expressions based on precedence value and size will, at every step of the search, contain at least one expression that precedes the target. Also, each new search round will produce slightly larger expressions until a solution is found. The parsimonious increase in size of preceding expressions then guarantees that a solution with minimum size will be found.

**The most similar to our work** that we know about is SR4MDL (Yu et al., 2025). More specifically, it also proposes learning a self-supervised heuristic model that guides the search for expressions afterwards. Like ours, their learning objective also comes from the structure of expressions and does not use fitness to data as a training signal. During training, the expressions also are synthetically generated at runtime. So far, we are not aware of other SR works that share those same characteristics. Still, there are important differences, as we show in Table 1.

Table 1: Main differences between our work and SR4MDL (Yu et al., 2025). The generative approach is also compared. [1] Minimum Description Length. [2] SME stands for the Sign-Mantissa-Exponent representation from (Kamienny et al., 2022).

| Aspect | Ours | SR4MDL | Generative |
|---|---|---|---|
| Objective | Formation precedence | MDL[1] | Fitness to data |
| Expression formation | Top-down | Bottom-up | Token sequence |
| Use | Dedicated to dataset | General purpose | Varies |
| Data assumptions | Just constants | Distribution of input | Varies |
| Training | Single phase | Two phases | Varies |
| Constant fitting | Any form | Limited forms | Any form |
| Input representation | Digit Transform (Eq. 3) | SME[2] | SME (common) |
| Search algorithm | Beam Search | MCTS | One shot, decoding |
| Optimal substructure | Yes | Yes | Does not apply |

**Datasets and benchmarks.** Possibly, the most well-known effort to standardize SR evaluation is SRBench (Cava et al., 2021). It contains more than 250 problems with and without ground-truth formulas. At least 14 methods have already been tested and compared (Makke & Chawla, 2024). SRBench includes the Strogatz (Strogatz, 2024) and Feynman (Feynman et al., 2011; Udrescu & Tegmark, 2020) dataset groups, the latter having some of the original physical laws removed. Other

Table 2: Example of primitives with the respective generation rules.

| Symbol | Rule |
|:---:|:---|
| $\square$ | $x \mapsto \square$ |
| $y$ | $x \mapsto y$ |
| $+$ | $x \mapsto +xx$ |
| $-$ | $x \mapsto -xx$ |
| $\cdot$ | $x \mapsto \cdot xx$ |
| $\sqrt{}$ | $x \mapsto \sqrt{x}$ |

Table 3: Expressions and respective prefix forms.

| Expression | Prefix Form |
|:---:|:---|
| $x + \sqrt{y}$ | $+x\sqrt{y}$ |
| $\square + x \cdot y$ | $+\square \cdot xy$ |
| $x - (y + \square)$ | $-x + y\square$ |

dataset groups for SR with ground-truth are available in (Keijzer, 2003; Vladislavleva et al., 2009; Uy et al., 2011; Korns, 2011; Petersen et al., 2021), but they are not physics-related.

**The reporting of SR results** still needs adherence to standardization. For instance, in (Biggio et al., 2021; Kamienny et al., 2022) authors report metrics based on $R^2 > 0.99$ as a proxy for symbolic solution on the Feynman problem subset from SRBench. As pointed out in (Matsubara et al., 2023), $R^2$-based accuracy does not take expression interpretability into account and is vulnerable to the use of dummy variables. Also, the $R^2$-criteria changes from work to work, sometimes being $R^2 > 0.9$ (d'Ascoli et al., 2024), $R^2 > 0.99$ (Kamienny et al., 2022), (Kamienny et al., 2023), (Shojaee et al., 2023b), while SRBench requires $R^2 > 0.999$. We stick to the Symbolic Solution Rate (SSR) defined in SRBench (Cava et al., 2021) as the main metric, but we still do use $R^2$ in the black-box datasets.

## 3 HTSSR: HEURISTIC BEAM SEARCH SYMBOLIC REGRESSION

Understanding the following components of our method is necessary for its comprehension. The basic constructs are the set of primitives and the generation rules. Then, expressions can be generated or randomly sampled with the rollout strategy. This generation procedure is at the core of the training data synthesis. That given, some care needs to be taken when evaluating the expressions numerically and feeding the heuristic neural network with such values.

### 3.1 PRIMITIVES AND GENERATION RULES

The mathematical expressions in this study are a combination of symbols, namely operators (unary and binary), variables, and constants. We call the set of all symbols the *primitives* set. Optionally, that set can be enriched with complexity constraints that tell the maximum allowed occurrences of a symbol under another symbol (e.g., at most 0 cosine operations inside a cosine). This controls the appearance of bizarre expressions and reduces the search space size. All considered expressions have a syntactical tree structure and are implemented using prefix notation. This choice of implementation allows for the fast generation, evaluation, and automatic differentiation of expressions.

Generation rules are defined in terms of the primitive symbols and their arities. One of the variables, $x$, is considered to be the special symbol used for rule applications. The generation rules have one of three forms: $x \mapsto o_2 x x$, $x \mapsto o_1 x$, and $x \mapsto o_0$. The $o_i$ indicate an operator with arity $i$. $o_0$ can be a variable name, including $x$, or the constant placeholder, $\square$. Multiple appearances of $\square$ represent independent constants. Tables 2 and 3 show examples of primitive sets, generation rules, and expressions with prefix forms.

### 3.2 EXPRESSION ROLLOUTS AND CANONICAL DATASET

Instead of working with a static dataset, we find it better to synthesize the expressions during the training of the heuristic model. The expressions are sampled in generation sequences, or *rollouts* (see Figure 1), where a source expression is first sampled from a static *canonical* dataset to then be expanded into increasingly more complex forms. This strategy gives access to a very large set of expressions, even when there are constraints for expression formation. (Kamienny et al., 2023) uses a mechanism similar to our rollouts in reverse order to generate expressions for training a mutation generative model. This model helps the main GP procedure in the search. Like our method, it

is a tree-search but uses MCTS instead of beam-search. Their method combines 3 parameterized models: a mutation policy, a selection policy, and a critic network. Ours, instead, only has one self-supervised model, trained for binary classification.

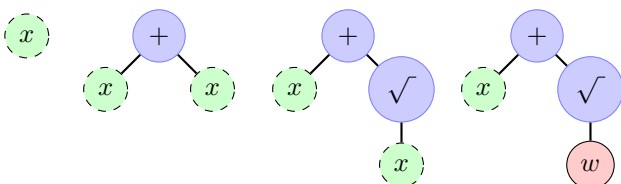

Figure 1: Example rollout from $x$ to $x + \sqrt{w}$. After the rollout is finished, $x$ becomes like any other variable.

The canonical dataset contains representatives of the numerical equivalence classes of expressions. The representatives are the smallest elements of a class. We define smallest as having the least number of primitive symbols and being the lexicographically smallest. If the constant placeholder is fixed, computing such a dataset and storing it on disk is possible up to some expression size. This limit also depends on the generation rules and on the primitives.

Uniformly sampling an entire set without considering complexity may underrepresent simpler expressions. We believe that such an imbalance makes the learning process harder. This is the main motivation behind the use of the rollouts. Regarding how the starting points of the rollouts are sampled, we see that sampling (uniformly on length) from canonical sets of different maximum expression lengths shows no significant difference (see Appendix A.2, Figure 12, for an ablation). However, the canonical set is important for the evaluation of the method, as the heldout datasets come from it.

### 3.3 NUMERIC EVALUATION

We make extensive use of stack-based evaluation of the expressions in prefix form. Given the limited scope of operations and the small number of variables, this solution is easy to implement and faster than SymPy (Meurer et al., 2017) and isolated Python code calls. The evaluation in the leaves involves variables and constants. The values attributed to the variables are the *feature domain* $\mathcal{D}[\mathbf{X}]$ - the part of the observed data $\mathcal{D}$ that is not the vector of target values $\mathcal{D}[\mathbf{y}]$. Constant placeholders are sampled from a uniform random distribution or get a fixed value. Our ablation in Appendix A.2, Figure 11, suggests that both choices result in very similar results. Operators get the result of being applied to their arguments. This happens until the top operation is computed.

The numeric results of expressions can easily get out of hand. Common problems are nondetermined (nan), overflow, underflow, and infinite values. To deal with values with large magnitude or that are infinite, we clip at a fake infinite (e.g. $\pm 10^{10}$). Overflows, underflows, and nondetermined results are avoided by the design of safe operators. For instance, a safe division attributes a floating-point number even if the result is not determined in the regular division. When the input domain is well behaved, the safe operators give the exact same results as the regular ones.

When performing prefix-order evaluation, there is a choice between keeping just the final result and also keeping the intermediary results of subexpressions. The last naturally distinguishes different expressions that have equal final values. The first needs some extra information for the distinction, like expression embeddings. We find that training with the first option converges with fewer iterations.

**Constant optimization**. The small number of numeric constants that might appear in the expressions works well with second-order optimization methods like Levenberg-Marquardt, taking between 4 and 12 iterations when converging. This is considerably faster than using first-order gradient methods like those based on SGD (Ruder, 2017). Using tools like Pytorch's *autograd* (Paszke et al., 2019), performing such inner optimizations is feasible. Because we implement all the evaluation processes, we can differentiate it with PyTorch.

## 3.4 THE HEURISTIC MODEL

Given an expression $E$ and the observed data $\mathcal{D}$, the heuristic models the probability that there is an expression $F$ such that $V(F)$, the evaluation of $F$, matches $\mathcal{D}[\mathbf{y}]$ and there is a rollout from $E$ to $F$. In other words, the heuristic tries to tell if a given expression is in the way of generating (or *precedes*) one expression that fits the data. The basic architecture (see Figure 2) has two parts: (i) an *encoder* that takes numeric values and outputs latent representations, and (ii) a binary classification module that takes a pair of outputs from the first module (plus some additional information about the potentially preceding expression) to predict the probability that one element precedes the other. The **Sort-Diff** and **Digit** transforms introduced in the following sections are performed in this order, before the parametric part of the encoder.

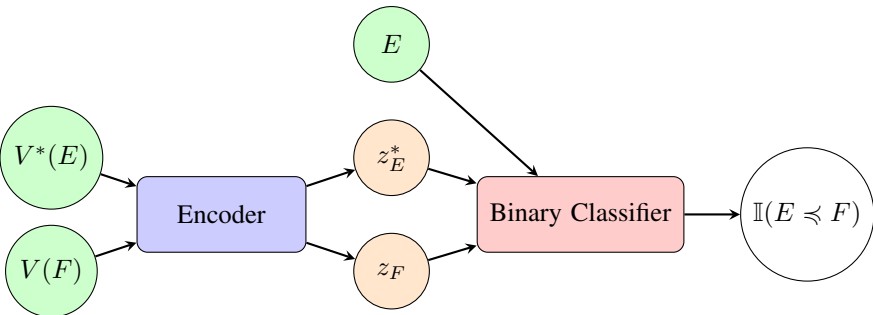

Figure 2: The generic form of the neural networks for the heuristic model. $E$ represents the potentially preceding expression, while $V(F)$ mocks the observed target data $\mathcal{D}[\mathbf{y}]$. The inputs to the encoder are processed independently. $V^*$ denotes the set of evaluations of all sub-expressions of $E$, including $E$ itself.

**Sort-Diff transform.** Motivated by the idea that information about the derivatives of the expression value with respect to input variables is helpful to learn the heuristic task, we introduce the Sort-Diff features. Those features consist of sorting $V(E)$ with respect to each input variable and then performing a diff operation on the sorted vector. This is supposed to be a surrogate for differentiation and can be applied to data that are not homogeneously sampled (e.g. there is no single step size). Notice that the observed data $\mathcal{D}$ cannot be automatically differentiated. The transformed evaluation vectors are concatenated with the original in a single vector. Equations 1 and 2 define the transform. We get better results when using Sort-Diff (see Appendix A.2, Figure 10 for an ablation).

$$\mathbf{Diff}(Y) := \{Y_{i+1} - Y_i\}_{0 \,\leq\, i \,<\, |Y|} \tag{1}$$

$$\mathbf{SortDiff}(Y, x) := \mathbf{Diff}(\{Y_i\}_{i \in \mathbf{ArgSort}(x)}) \tag{2}$$

**Digit transform.** It is known that having high differences in value ranges from feature to feature affects the stability and convergence of optimization during training. Because expression evaluations in SR do suffer from such differences in range, we introduce a transformation that, for every single number in the input data, outputs a vector. This vector contains what would be digits in a base $b$ representation. For a suitable value of $b$, each input feature can have a standardized and optimization-friendly range. Equation 3 defines the transform. Here, $a\%b := a - \lfloor a/b \rfloor \cdot b$.

$$\mathbf{DigitTransform}(x) := (x \cdot b^{[-d, -d+1, \ldots, d]}) \,\%\, b \tag{3}$$

Common normalization techniques like Min-Max and Mean-Std lose scale information, which is fundamental for the SR task. Transformations that try to make high values more amenable, like taking the logarithm, might squeeze values from higher ranges into smaller intervals, making their representations less useful. The input representation introduced in (Kamienny et al., 2022) and commonly used by other works like (Yu et al., 2025), (Shojaee et al., 2023a) is token-based and consists of three tokens: a sign, a mantissa, and an exponent. The mantissa represents 4 significant positions using the tokens from 0 to 9999. The exponent tokens range from $-100$ to $100$ and

therefore the total of distinct input representations is in the order of $10^8$. On the other hand, our digit transform can give a representation for every floating point number, with roughly 7 significant positions for 32-bit float and 16 for 64-bit float. Because SR explores a combinatorially vast space, the more expressive digit transform can improve performance. We provide a comparison between our digit transform and an adapted version of the sign-mantissa-exponent representation in Appendix A.2, Figure 9.

**Binary classifier.** Each pair of outputs from the encoder can be combined in different ways before entering the classifier. In our experiments, the best approach was to take the difference between the latent representations and then add positional encodings and expression embeddings. Since precedence is antisymmetric, subtracting (not adding) the latent vectors better distinguishes input order. The loss function is the binary cross-entropy.

**Training with all-pairs mini-batches.** During training, a set of rollouts is sampled such that the starting points have an equal chance of having any length from 1 to the maximum length of the canonical set. Only starting points are guaranteed to not have a smaller form, up to simplification of constant sub-expressions. Then, when the collection of rollouts reaches a certain number of expressions (e.g., 32), the binary labels (precedes or not) are computed for all ordered pairs of elements. It is easy to do that for pairs of the same rollout, as the expressions that appear first precede the ones that appear later. For pairs of different rollouts, the syntactic trees are compared. We use the convention that any expression precedes itself.

**Synthetic heldout datasets**. For each number of variables $n_{var} \in \{1, 2, 3, 4\}$, a set with 30 expressions for each expression length fom 5 to 10 is created (except for 4 variables, which require at least 7 symbols). Each expression is sampled from the canonical set created with the respective number of variables, but keeping the rest of primitives the same. Unlike rollouts, this sampling is uniform given the number of variables and length. Also, cases where an expression simplifies to a simpler one only happen when the canonical expression has a subexpression of composite constants (e.g. $\square \cdot e^{\square}$). When evaluating on these heldout sets, expressions that simplify are counted as having the shorter length. Check Appendix A.10 to see the heldouts.

### 3.5 BEAM SEARCH

The search starts from $x$ and keeps creating new expressions by expanding leaf nodes with $x$. These expansions are exhaustive: for each combination of $x$ leaf and generation rule, a new expression is formed. It uses the same set of primitives and generation rules used to train the heuristic model. Then, each expression is numerically evaluated and fed to the heuristic model. Then, a priority queue receives the expressions with their respective priorities. Whenever an expression without constants (purely operators and variables) is taken from the queue, it is evaluated and compared to the observed values. If the relative squared error is less than some threshold (e.g. $10^{-4}$), it returns the solution. In case the expression has at least one placeholder for constants, a subroutine optimizes for the constants and, if converging, returns the parameter values. The main routine then applies the same acceptance criteria. If a maximum number of expressions is visited, the search stops. The pseudo-code for HTSSR is available in Appendix A.5, Algorithm 1. The acceptance criterion is defined in terms of a relative tolerance and the relative squared error between the target $\mathbf{y}$ and the expression evaluation $\hat{\mathbf{y}}$:

$$RSE(\mathbf{y}, \hat{\mathbf{y}}) := \frac{\sum (y_i - \hat{y}_i)^2}{\sum y_i^2} \qquad (4)$$

## 4 EXPERIMENTS

Next, we first analyze HTSSR with respect to dataset size and scalability (Section 4.1). Then, we show the results of HTSSR on the ground-truth (Feynman and Strogatz) and black-box dataset groups from SRBench (Section 4.2). The scalability experiments show how the Symbolic Solution Rate (SSR) changes given expression length and some other aspects, which are the number of variables and the number of domains $\mathcal{D}$. The evaluation datasets used in Section 4.1 are the same heldouts described in Section 3.4 and are integrally shown in Appendix A.10. The default domain is

`feynman_I_34_1`, appearing other domains only in the SRBench and domain scalability experiments. More details about configuration can be found in Appendices A.6 A.7.

## 4.1 EFFICIENCY ON DATASET LENGTH AND SCALABILITY

**Efficiency on dataset length**. We investigate how search performance changes when changing the availability of data points. The results in Figure 3 support the idea that, under similar conditions, more data points produce better results. Increasing one order of magnitude from $10^2$ to $10^3$ data points shows little to no gain, while increasing from $10^1$ to $10^2$ shows clear gains. Importantly, HTSSR can find solutions with as few as 10 points, which supports the idea that the method has potential in a data-scarce scenario. The *dummy* baseline shows the brute force of beam search, where the heuristic is clueless but still can find some simple expressions under the imposed conditions. The dummy trial also makes this experiment into an ablation of the heuristic model, indicating that the model does make a difference.

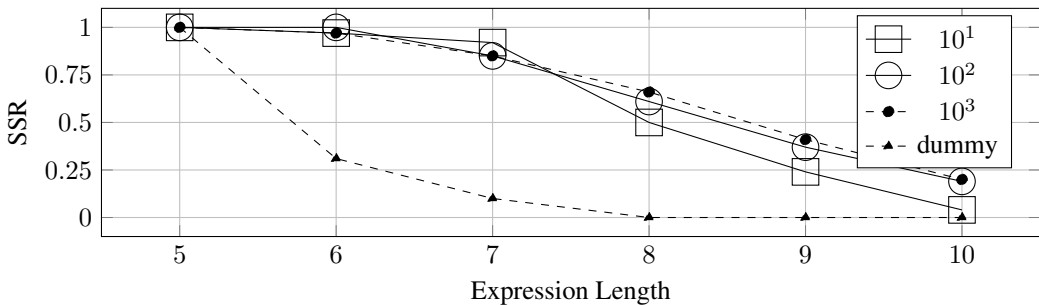

Figure 3: SSR vs. expression length for sample sizes $10^1$, $10^2$, $10^3$, and a dummy model. $n_{var} = 3$.

**Scalability: number of variables.** Now we investigate the impact that $n_{var}$ has on the SSR. From Figure 4, it looks like the expression length plays a more important role in the decay of the SSR than the number of variables. Only for $n_{var} = 4$ versus $n_{var} < 4$ is there a clear sign of degradation for expression length greater than 7. Furthermore, it seems that the decay of SSR for $n_{var} = 1$ is slower at larger lengths. It could be that for $n = 1$ it is possible to find solutions larger than 10 symbols somewhat frequently. We invite the reader to look at the complementary scalability analysis in Appendix A.3.

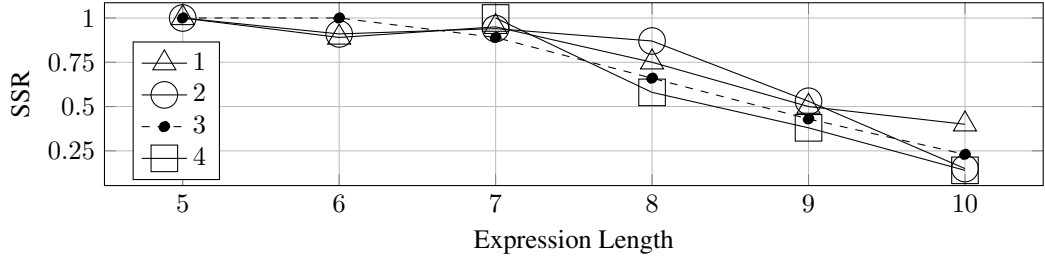

Figure 4: SSR versus expression length for $1 \leq n_{var} \leq 4$.

**Scalability: number of domains**. Figure 5 indicates that increasing the number of domains in which a single heuristic model is trained (using learnable domain embeddings) degrades its quality on the evaluation domain `feynman_I_34_1`, also seen during training, at least for $n_{var} = 1$ versus $n_{var} > 1$. However, among $n_{var} > 1$, the degradation of SRR is relatively small, if any. This might indicate potential for reusability of the heuristic model, as one single model could be used for many data domains.

## 4.2 SRBENCH

**Feynman and Strogatz datasets.** We run HTSSR on the Feynman (119 datasets) and Strogatz (14 datasets) dataset groups under the constraints of SRBench for ground-truth problems. There

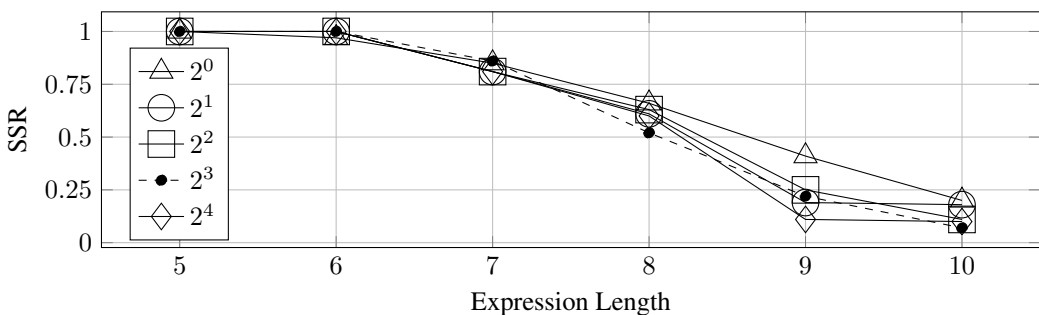

Figure 5: SSR versus expression length for 1 to 16 domains in the same heuristic model. $n_{var} = 3$.

are training time limits of 36000 and 3600 seconds for each problem in Feynman and Strogatz, respectively. Within the training time budget, model checkpoints at different epochs are used to search.

HTSSR ranks among the top methods and its performance drops mainly at 0.1 noise in Strogatz, less so in Feynman. The only noise level that visibly disturbs our method is 0.1. Except for SR4MDL (on both dataset groups) and AIFeynman2 (Udrescu et al., 2020) (on the Feynman datasets), HTSSR with 0.1 noise level surpasses or is equivalent to the other methods with 0.0 noise. In addition, the performance of HTSSR is consistent when changing dataset groups, as it does not make specific assumptions about the problems. In principle, HTSSR could score higher if helped with problem simplification or a divide-and-conquer approach, where a problem is decomposed into sub-problems. Also, variations of HTSSR that replace the search algorithm are worth investigating.

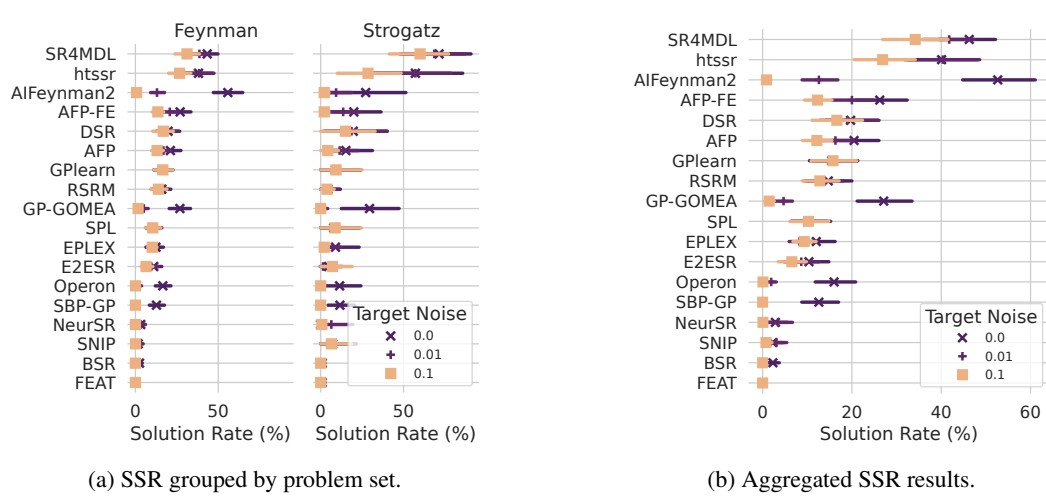

(a) SSR grouped by problem set.

(b) Aggregated SSR results.

Figure 6: Comparison with the SRBench results of other methods.

**Black-box datasets.** The black-box results (122 datasets) reveal that HTSSR is able to find very concise and moderately accurate results. It appears on the Pareto front of the $R^2$ versus model size (see Figure 7) and dominates the generative methods E2ESR (Kamienny et al., 2022) and NeuralSR (Biggio et al., 2021). It performs similarly to DSR (Petersen et al., 2021), but using less time, Figure 8. Consider that DSR also needs to train from zero for every new dataset. In $R^2$ alone, HTSSR is closer to MLP, but has more than two orders of magnitude smaller sizes. The conceptually close SR4MDL (Yu et al., 2025) has around 25% higher $R^2$, but almost $10\times$ larger expressions.

## 5  CONCLUSION

This paper presents a new and simple method for SR with key advantages, making a shift from common approaches in the literature that use fitness to data as training signal or that explore the

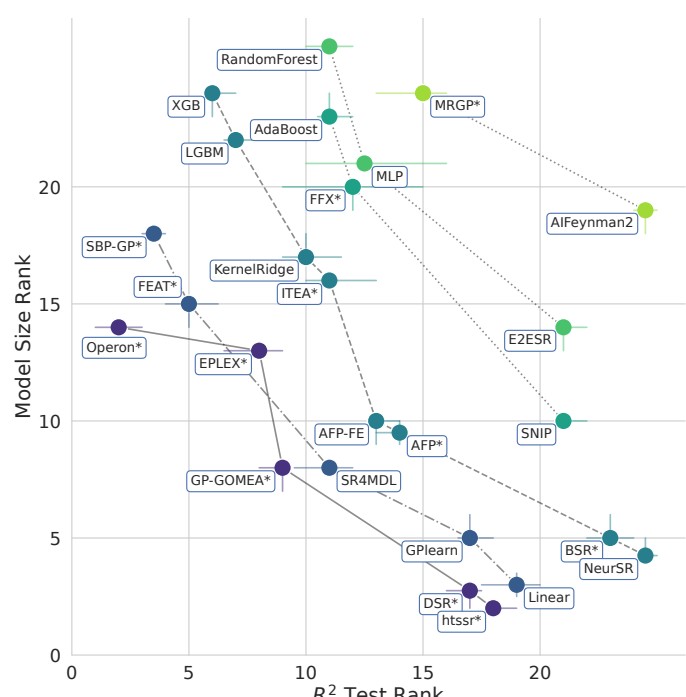

Figure 7: Pareto plot of results on the black-box datasets.

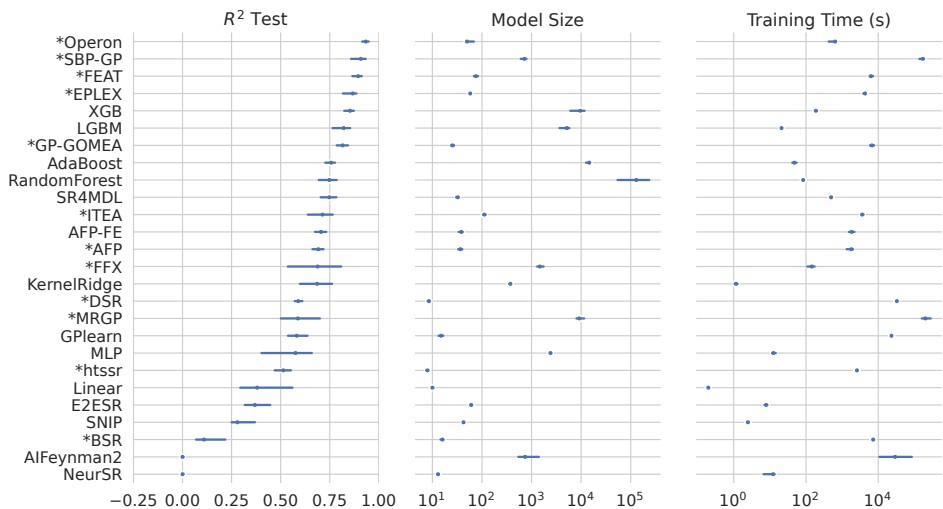

Figure 8: $R^2$, model size, and training time on the black-box datasets.

space of solutions on a token-by-token manner. It finds solutions with desired properties, such as exactness, and simplicity, while being competitive with existing methods and less affected by noise. We also analyze some aspects of scalability and efficiency in data scarcity, providing insight into further investigations and improvements. We find that the major factors that affect the effectiveness of the method seem to be the length of expressions and the number of data points. However, the clean design and modular nature of the method is encouraging for adaptations and developments.

## 6 REPRODUCIBILITY

We plan to soon release a refactored version of the code and instructions to the public. As of now, code and instructions are available as suplementary material for the reviewers in the reviewing platform. Each experiment ran on a NVIDIA A100-80GB GPU with single process at a maximum 2.2 GHz processor core.

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

# A APPENDIX

## A.1 LLM USAGE

In this work, LLMs helped spot typos and suggest words in a few cases.

## A.2 ABLATIONS

**Digit Transform**. Figure 9 compares the performances of the Digit Transform and SME input representations. There is a clear pattern of dominance of Digit Transform over the expression lengths.

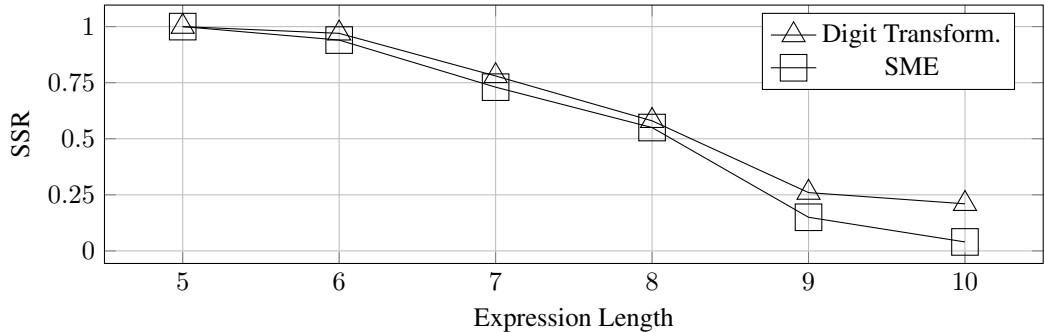

Figure 9: Comparison between our digit transform and the sign-mantissa-exponent (SME) representation. We adapted the SME version to comply with our numeric encoder such that both neural networks have the same size specifications.

**Sort-Diff**. In Figure 10 there is a clear pattern that shows the superiority of applying the Sort-Diff transform to input features versus not. The results show dominance of Sort-Diff across all expression lengths.

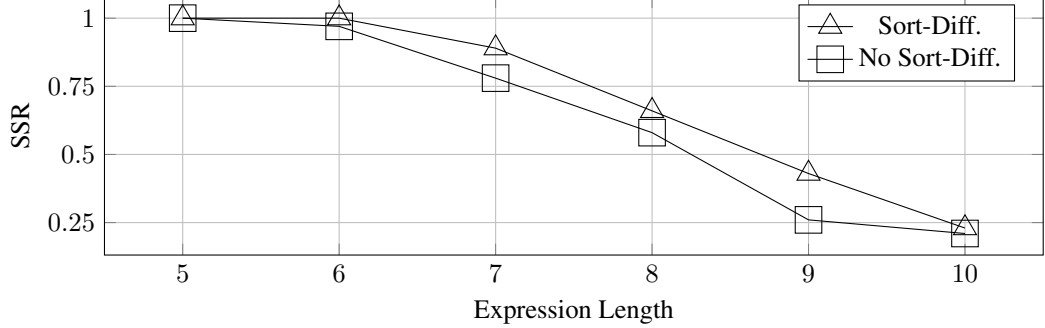

Figure 10: Symbolic Solution Rate (SSR) versus expression length for model with and without the **SortDiff** transform. $n_{var} = 3$.

**Evaluation of the constant placeholder**. Figure 11 shows very close tendencies when comparing the SSR resulting from heuristics trained with a fixed value $v_\square$ versus the sampled value $v_\square + U(-0.1, 0.1)$. The motivation behind this experiment is to see if sampling $\square$ improves the ability of the heuristic model to perform well for expressions with constants that are not seen during training. The results have only small, opposite differences at the lengths 9 and 10 and suggest that no difference is revealed.

**Maximum size in the canonical set**. In Figure 12 there is a comparison between the SSRs resulting from heuristics trained by sampling the starting points of rollouts from canonical datasets of different sizes. The idea of using canonical datasets to anchor the sampling is that it would make the mini-batches more balanced with respect to expression length. This was expected to yield better heuristics, but the results show no improvement. In part, this could be because the rollouts naturally

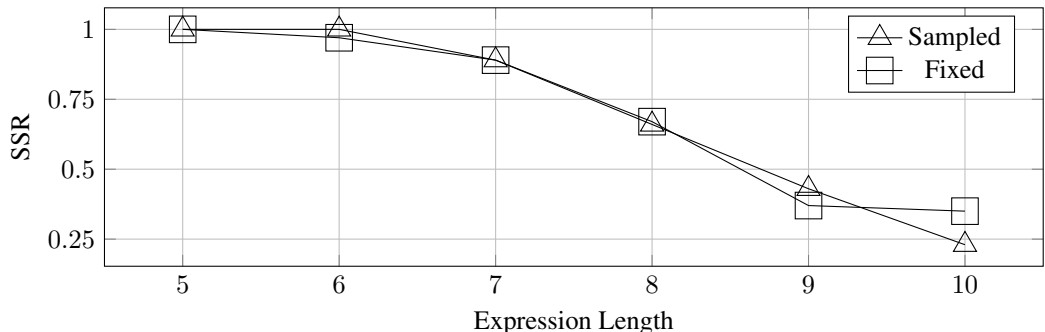

Figure 11: Symbolic Solution Rate (SSR) versus expression length for □ sampled versus fixed during training. $n_{var} = 3$.

create expressions with varying complexities, and the expressions that simplify are not sufficient to impact the representation of larger expressions negatively. On the other hand, the larger number of longer expressions do not affect the representation of smaller ones because of the nature of rollouts.

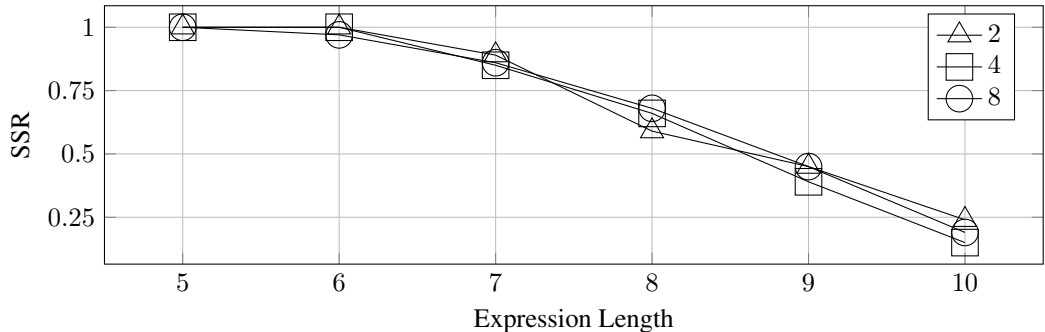

Figure 12: Symbolic Solution Rate (SSR) versus expression length for canonical sets with maximum expression lengths 2, 4, and 8. $n_{var} = 3$.

### A.3 COMPLEMENTARY SCALABILITY ANALYSIS

In order to have a broader idea about the power of the proposed heuristic, we perform simulations where a simulated heuristic model is characterized by two parameters: the Recall at the positive and negative classes. This is possible by taking the ground-truth precedence signal and flipping it with some probability. We combine all pairs of Recall@0 and Recall@1 from the set $\{0.75, 0.80, 0.85, 0.90, 0.95\}$ for each of two search algorithms: the stochastic search (repeated random rollouts based on probabilities) and the beam search (with fixed beam window of 32). We also repeat for an additional parameter: the maximum allowed size of an expression in the search, which can be 12 or 18. Every search run can visit up to $2 \cdot 10^4$ different expressions and an expression can be visited multiple times counting as one. The result of each search is either the solution or nothing.

The expressions to be found are a subset of the expressions in the Feynman dataset. There are 73 of size up to 12 and 96 with size up to 18. The number of variables ranges from 1 to 6. The distribution of expression sizes can be seen in Figure 13 and the distribution of number of variables in Figure 14. The set of primitives is the set in Table 4, but adding the operators arccos and arctan.

The aggregated SSR results are shown in Figure 15. We see that for beam search the Recall@1 is a more important factor in success than Recall@0 and that the algorithm tends to get lost if not pruned, as the results with maximum size 12 are much higher than those for 18. The results for beam search start to get better when Recall@1 is around 0.95, if properly pruned. On the other hand, the stochastic search seems less sensitive to the maximum allowed size and its quality is affected by both Recall@0 and Recall@1 more equally.

Figures 16 and 17 shows how the SSR varies along expression lengths for different simulated heuristic performances. While the stochastic search can find longer expressions and improves gradually with the recalls, the beam search indeed seems to require pruning and a high value of Recall@1 to perform well in the shorter range. The variations with respect to $n_{var}$ are shown in Figures 18 and 19. Knowing that more variables mean longer expressions, it is natural to expect a decrease in SSR when the number of variables increases.

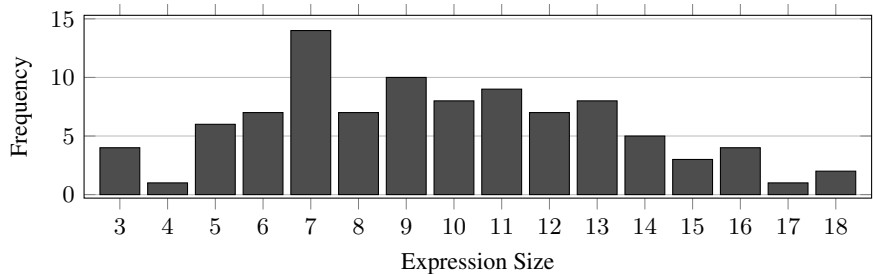

Figure 13: Frequency of expression sizes in the chosen subset for the simulated heuristics.

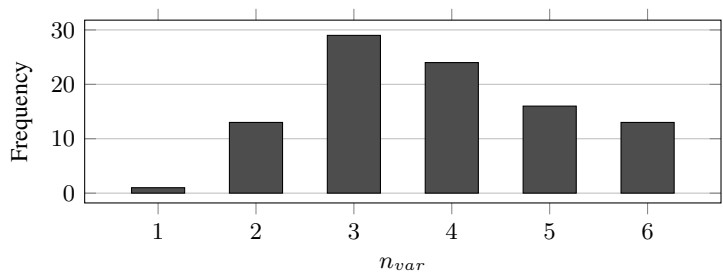

Figure 14: Frequency of $n_{var}$ in the chosen subset for the simulated heuristics.

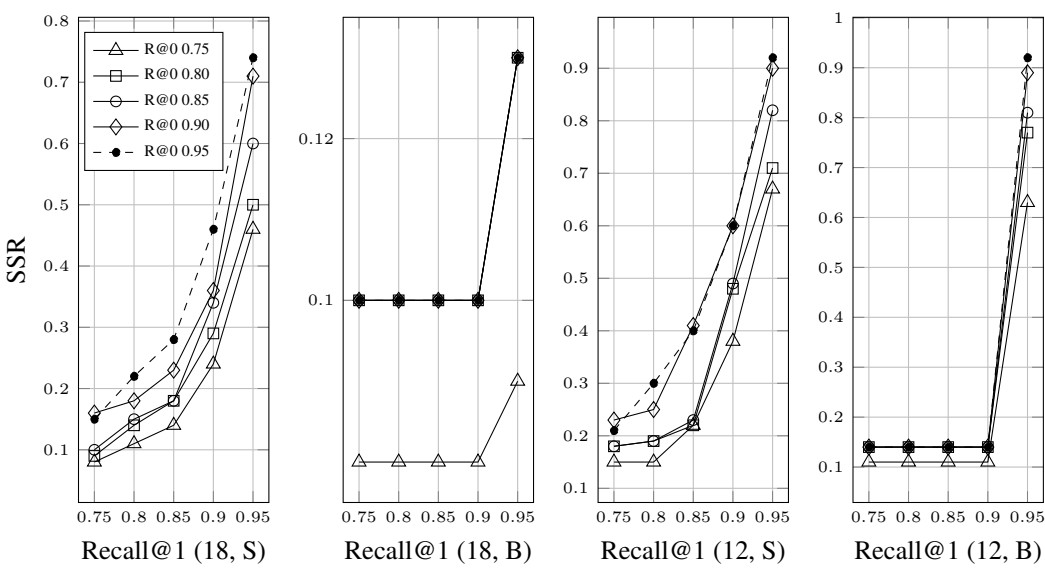

Figure 15: SSR versus Recall at the positive (precedes) class for each Recall at the negative class (does not precede). The (12/18, S/B) annotations indicate two parameters: maximum allowed expression size in the search and if the search algorithm is the stochastic search or the beam search.

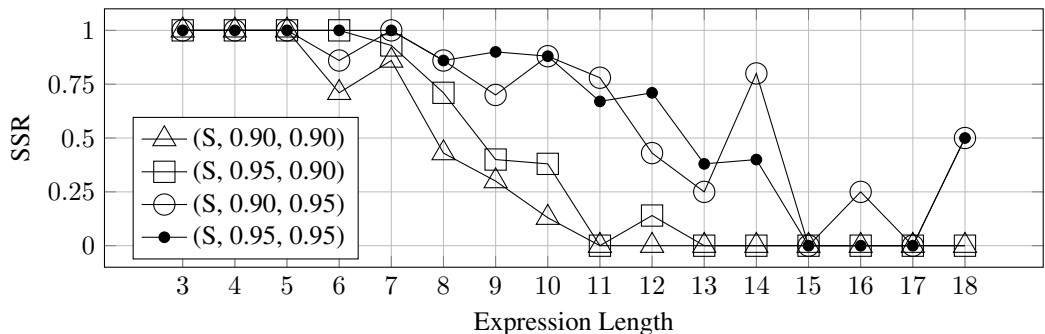

Figure 16: SSR versus expression length for the stochastic search (S) with the simulated heuristic with different Recall@0 and Recall@1.

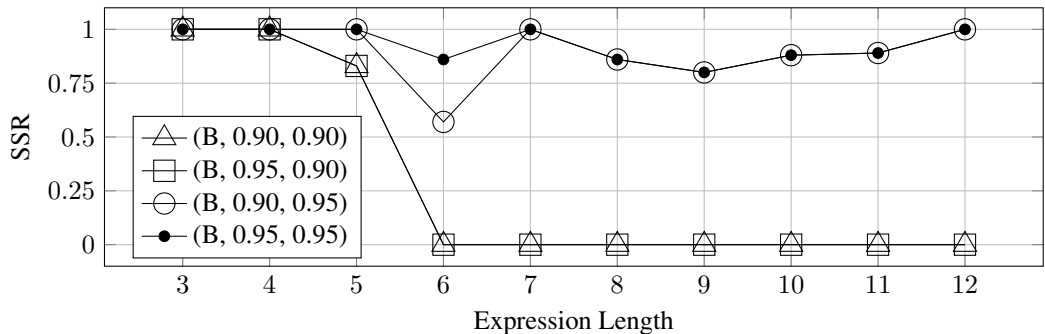

Figure 17: SSR versus expression length for the beam search (B) with the simulated heuristic with different Recall@0 and Recall@1.

## A.4 SET OF PRIMITIVES

Table 4 shows the set of primitive symbols. Table 5 shows the constraints used in the formation of expressions for the experiments.

Table 4: Constants, variables, and operators used in the experiments. arcsin was used only in the SRBench experiment (Section 4.2).

| Symbol | $\square$ | $x$ | $y$ | $z$ | $w$ | $+$ | $-$ | $\cdot$ | $/$ | $.^2$ | $\sqrt{\phantom{x}}$ | sin | cos | $e^{\cdot}$ | arcsin |
|--------|-----------|-----|-----|-----|-----|-----|-----|---------|-----|-------|------------|-----|-----|-----|--------|
| Arity | 0 | 0 | 0 | 0 | 0 | 2 | 2 | 2 | 2 | 1 | 1 | 1 | 1 | 1 | 1 |

## A.5 ALGORITHMS

Algorithm 1 synthesizes the high-level workings of the beam-search, given a trained heuristic $h_\Theta$. Algorithm 2 is a simplified version of the implementation for creating canonical sets of expressions.

## A.6 SETUPS FOR THE SCALABILITY AND SAMPLE EFFICIENCY EXPERIMENTS

Every heuristic model in that part of the experiments was trained for 1000 epochs of 50 iterations each. The mini-batches were all-pairs of size $16 \times 16$. The beam search window is 128 and the limit of visited states is 10240. Except for the multi-domain experiment, the default domain used is from the problem `feynman_I_34_1`, with the extra fourth variable being sampled from $U(1, 5)$. $\mathcal{D}$ is randomly sub-sampled from $10^5$ to $10^3$ data points (and to $\{10^2, 10^1\}$ in Section 4.1). Details about the neural net configuration are in Appendix A.8.

---

**Algorithm 1** HTSSR, based on beam-search.

---

$Q \leftarrow [(0, x)]$                                                ▷ Initialize priority queue
$V \leftarrow \{ \}$                    ▷ Set of visited states
**while** $length(V) \leq m$ **do**        ▷ Maximum of $m$ visited states
     **if** $length(Q) = 0$ **then**
         **return**           ▷ No solution found
     **end if**
     $B \leftarrow [Q.pop(), ..., Q.pop()]$    ▷ Beam size pops while not empty
     **for** $s, E \in B$ **do**    ▷ Iterate through priority-expression pairs
         **if** $\square \notin E$ and $Accept(Eval(E), \mathcal{D})$ **then**
             **return** $E$       ▷ Constant-free solution found
         **else if** $\square \in E$ **then**
             $\xi \leftarrow LM(E, \mathcal{D})$    ▷ Run Levenberg-Marquadt optimization
             **if** $Accept(Eval(E), \mathcal{D}, \xi)$ **then**
                 **return** $E, \xi$      ▷ Solution with constant(s) found
             **end if**
         **end if**
         $C \leftarrow Expand(E)$      ▷ Get the set of children expressions
         $S \leftarrow 1 - \sigma(h_\Theta(C, \mathcal{D}))$    ▷ Attribute priority scores with the learned heuristic, $h_\Theta$
         $Q.push((S, C))$      ▷ Update the priority queue
         $V.add(E)$
     **end for**
**end while**

---

**Algorithm 2** Creation of canonical set of expressions up to length $n$.

---

$S \leftarrow O_0$                  ▷ Initialize canon set with zero-ary elements.
$V \leftarrow \{ \}$                 ▷ Initialize visited values.
**for** $2 \leq l \leq n$ **do**        ▷ Iterate from lengths $2$ to $n$.
     **for** $o \in O_1$ **do**       ▷ For each unary operator
         **for** $F \in S_{l-1}$ **do**    ▷ For each expression in $S$ with length $l - 1$
             $E \leftarrow o(F)$      ▷ Create new expression of length $l$
             **if** **Eval**$(E) \notin V$ **then**    ▷ Add only if a smaller one is not equivalent.
                 $S.append(E)$
                 $V.add(\mathbf{Eval}(E))$
             **end if**
         **end for**
     **end for**
     **for** $o \in O_2$ **do**          ▷ For each binary operator
         **for** $1 \leq l^L \leq n - 2$ **do**      ▷ For each length of the left subtree
             $l^R \leftarrow n - 1 - l^L$
             **for** $F^L \in S_{l^L}$ **do**
                 **for** $F^R \in S_{l^R}$ **do**
                     $E \leftarrow o(F^L, F^R)$
                     **if** **Eval**$(E) \notin V$ **then**
                         $S.append(E)$
                         $V.add(\mathbf{Eval}(E))$
                     **end if**
                 **end for**
             **end for**
         **end for**
     **end for**
**end for**
**return** $S$

---

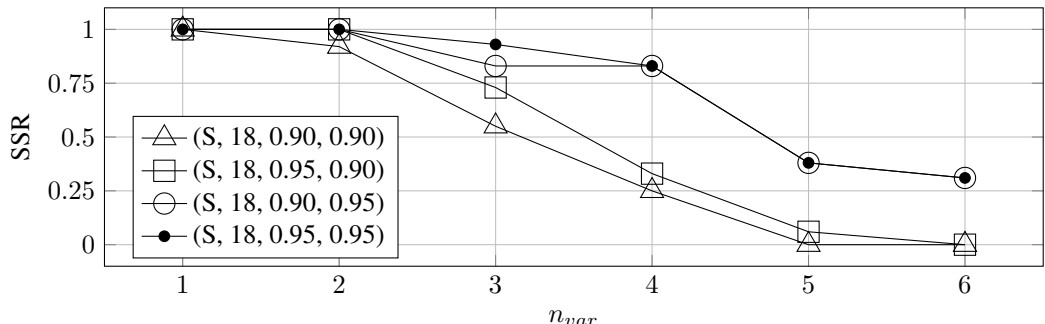

Figure 18: SSR versus $n_{var}$ for the stochastic search (S) with the simulated heuristic with different Recall@0 and Recall@1.

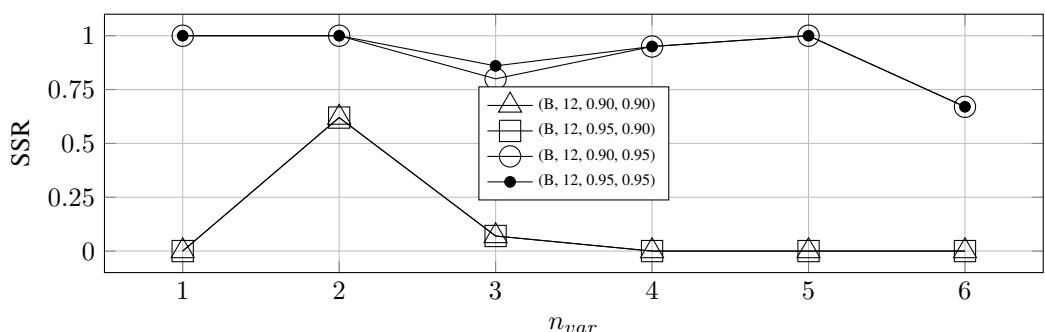

Figure 19: SSR versus $n_{var}$ for the beam search (B) with the simulated heuristic with different Recall@0 and Recall@1.

## A.7 SETUPS FOR THE SRBENCH EXPERIMENT

The general process for searching for a solution of a given problem starts by training the heuristic model. Training is interrupted at defined epochs ([2499] for Strogatz and [2499, 9999, 25999] for Feynman) so that HTSSR uses the current checkpoint to search. The beam of the search is 16384. The limit of visited states is 102400. The relative tolerance to accept a candidate solution and stop the search is $2 \cdot 10^{-4}$. If the threshold is not met but there is still time remaining, the checkpoint goes back to training. The search ends if the threshold is met or if time is out, in which case the expression with the lowest relative error is returned. Other configuration and neural net structure are described in Appendix A.8.

## A.8 NEURAL NET ARCHITECTURE

Table 6 shows the main neural net configuration used across experiments. The main difference between experiments is at the first layer, as the number of input units is different between dataset groups (10, 100, 1000 for Feynman, 300 for Strogatz). In the Self-Attention layers, $d_{model} = 1024$ for all experiments except for the SRBench experiment, where $d_{model} = 768$. The "Linear" layers in the numerical encoder have standard 2048 hidden-layer width, with final layer width being $d_{model}$. The exception is for the SRBench experiment, where those hidden Linear layers have width 1024. In the Digit Transform, all experiments use 67 digit equivalents in base 2, with position values from $2^{-33}$ to $2^{33}$.

## A.9 TRAINING TIMES

Figure 20 compares the training times of HTSSR and two methods with similar performance: DSR and BSR. Here, the most relevant comparison is between HTSSR and DSR, as both are learning-based and dedicated to each given dataset and need to be trained from zero.

Table 5: Constraints for the formation of expressions. Row elements can appear up to the specified number of times under the column element in the expression syntactic tree. Empty cells indicate no constraint.

|  | $+$ | $-$ | $\cdot$ | $/$ | $.^2$ | $\sqrt{\ }$ | sin | cos | $e^{\cdot}$ | arcsin |
|---|---|---|---|---|---|---|---|---|---|---|
| $\square$ | 2 | 2 | 2 | 2 | 2 | 2 | 2 | 2 | 2 | 2 |
| $x$ |  |  |  |  |  |  |  |  |  |  |
| $y$ | 2 | 2 | 2 | 2 | 2 | 2 | 2 | 2 | 2 | 2 |
| $z$ | 2 | 2 | 2 | 2 | 2 | 2 | 2 | 2 | 2 | 2 |
| $w$ | 2 | 2 | 2 | 2 | 2 | 2 | 2 | 2 | 2 | 2 |
| $+$ |  |  |  |  |  |  |  |  |  |  |
| $-$ |  |  |  |  |  |  |  |  |  |  |
| $\cdot$ |  |  |  |  |  |  |  |  |  |  |
| $/$ |  |  |  |  |  |  |  |  |  |  |
| $.^2$ |  |  |  |  | 0 | 0 | 0 | 0 | 1 | 0 |
| $\sqrt{\ }$ |  |  |  |  |  | 0 | 0 | 0 | 0 | 0 |
| sin |  |  |  |  |  |  | 0 | 0 | 0 |  |
| cos |  |  |  |  |  |  | 0 | 0 | 0 | 0 |
| $e^{\cdot}$ |  |  |  |  |  |  | 0 | 0 | 0 | 0 |
| arcsin |  |  |  |  |  |  |  |  |  | 0 |

Table 6: General Neural Net Configuration for the Experiments.

| Module | Submodules |
|---|---|
| Numeric Encoder | SortDiff (optional) |
|  | Digit Transform |
|  | $3\times$ (Linear, RMSNorm, GELU) |
| Source-Target aggregation | $-$ (difference) |
|  | Final result or all-tree results |
| Positional Encoding | $+$ (padded to length 15) |
| Positional Encoding (parent symbol) | $+$ (optional) |
| Expression Embeddings | $+$ (optional) |
| Domain Embeddings | $+$ (optional) |
| Classification | $4\times$ Self-Attention (4 heads) |
|  | Sequence aggregation (mean) |
|  | Linear |

## A.10 EXPRESSION HELDOUTS

The following Tables 7 8 9 10 11 12 13 14 contain the heldout dataset groups used in the experiments (except SRBench). Those are grouped by $n_{var}$.

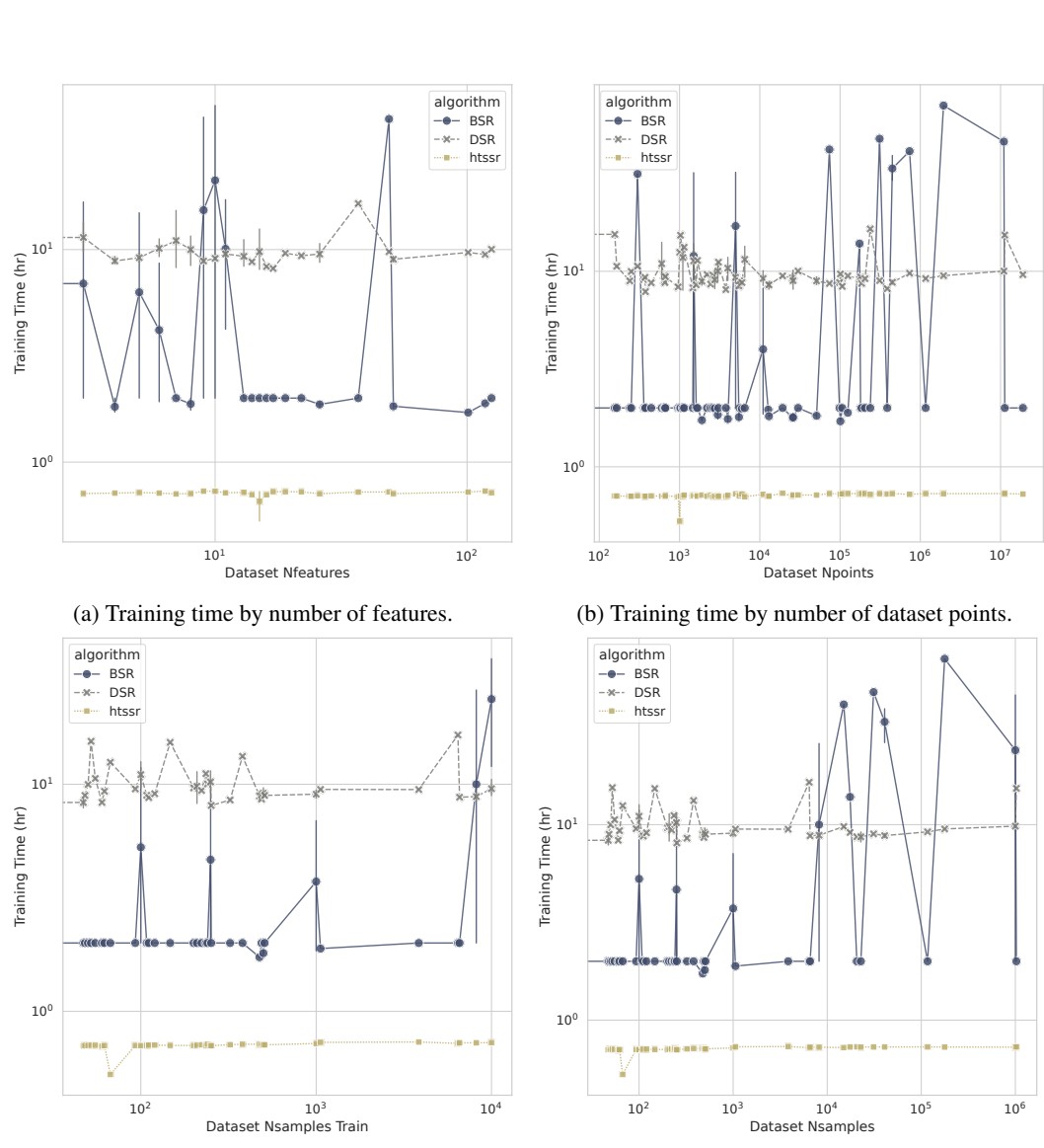

(a) Training time by number of features.

(b) Training time by number of dataset points.

(c) Training time by number of training samples.

(d) Training time by number of total samples.

Figure 20: Training times comparisons with methods that perform similarly.

Table 7: Heldout expressions for $n_{var} = 1$. Part 1.

| | | |
|---|---|---|
| $(e^x - \sqrt{\Box})$ | $(x \cdot \sqrt{\sin(x)})$ | $((x - \cos(\Box)))^2$ |
| $(\sqrt{\sin(\Box)} - x)$ | $((\Box + \sqrt{x}))^2$ | $\frac{(x-\Box)}{\frac{\Box}{(\cos(x))^2}}$ |
| $\sqrt{\sin((\Box - x))}$ | $(\cos(\Box) - (x)^2)$ | |
| $\frac{x}{(e^x)^2}$ | $\frac{\sqrt{e^x}}{\Box}$ | $\frac{x}{\sqrt{\sin(x)}}$ |
| $\frac{\Box}{\sqrt{(x)^2}}$ | $\frac{e^x}{\sin(x)}$ | $((x \cdot \sin(\Box)))^2$ |
| $(e^{(\Box \cdot x)} - x)$ | $((x + \frac{x}{\Box}))^2$ | $(x + \cos((\Box \cdot x)))$ |
| $\frac{x}{(\Box + e^\Box)}$ | $(x \cdot \cos((\Box \cdot x)))$ | $(\Box + \frac{\Box}{\sin(x)})$ |
| $\frac{(\Box - x)}{\sqrt{\Box}}$ | $\sqrt{(\sin(\Box) - e^x)}$ | $(\Box + ((x)^2 - x))$ |
| $(\sqrt{\frac{x}{\sin(x)}})^2$ | $((\Box)^2 - \sqrt{\sin(x)})$ | $(\sqrt{(x)^2} \cdot e^\Box)$ |
| $\frac{(x-\Box)}{\cos(\Box)}$ | $\frac{\Box}{(\Box + e^x)}$ | $(e^x - \sqrt{\sin(\Box)})$ |
| $(x \cdot \frac{\cos(\Box)}{e^x})$ | $(x - \frac{e^x}{\cos(\Box)})$ | $(\sin(\frac{\Box}{x}) + \cos(\Box))$ |
| $(\frac{\sin(x)}{\Box} - e^x)$ | $((e^\Box)^2 - \frac{\Box}{x})$ | $\frac{\Box}{\sqrt{((x)^2 - \Box)}}$ |
| $(\cos(x) - \frac{x}{\sin(x)})$ | $(\Box + (x + (e^\Box)^2))$ | $(e^{(\Box + x)} - \sin(x))$ |
| $(x - \frac{\sqrt{\cos(x)}}{\Box})$ | $(\Box \cdot \sqrt{(\Box \cdot (x)^2)})$ | $\frac{\sqrt{x}}{((x)^2 - \Box)}$ |
| $((x \cdot \sqrt{\Box}) - \sqrt{x})$ | $\sqrt{(\Box \cdot (x + \sin(\Box)))}$ | $\frac{e^\Box}{(\Box - \sqrt{x})}$ |
| $(\Box \cdot \sqrt{(\sin(x) + e^x)})$ | $(\sqrt{\sin(\Box)} - \frac{x}{\cos(\Box)})$ | $((\frac{x}{\cos(x)} + e^\Box))^2$ |
| $\frac{\frac{(x)^2}{\sqrt{\Box}}}{\cos(\Box)}$ | $(\cos(x) - \frac{\Box}{(\sin(\Box))^2})$ | $\sqrt{(\sin(\Box) - \sin((\Box \cdot x)))}$ |
| $(\sqrt{(\frac{x}{\cos(\Box)} - \Box)})^2$ | $\sqrt{\frac{\Box}{(\cos(\frac{x}{\Box}))^2}}$ | $(\frac{\sin((\Box + \Box))}{\sqrt{x}})^2$ |
| $((\frac{\Box}{x} + \frac{x}{\Box}))^2$ | $\frac{(e^x - (\cos(\Box))^2)}{\Box}$ | $\frac{x}{((\Box + x) \cdot \sqrt{\Box})}$ |
| $\sqrt{(\Box \cdot \frac{\cos(x)}{\sin(x)})}$ | $(x \cdot \sin((\Box - (\Box \cdot x))))$ | $(\frac{\Box}{\cos(x)} + (\sqrt{x})^2)$ |
| $(\sqrt{x} + \cos(\frac{x}{(x+x)}))$ | $\frac{(\cos(\Box))^2}{(\Box + (\sqrt{x})^2)}$ | $((\frac{x}{(x+x)} - \sqrt{x}))^2$ |
| $(((\Box + x) \cdot e^\Box) + \cos(x))$ | $(\Box \cdot \sqrt{(e^{\frac{\Box}{x}} - x)})$ | $(((x + \sqrt{\sin(\Box)}) \cdot \sqrt{x}))^2$ |
| $(((\sin(\Box))^2 - \sin(x)) - \cos(x))$ | $(x - \frac{(x + \sqrt{\Box})}{\sin(x)})$ | $(x + (x + (x - \sqrt{\sin(\Box)})))$ |
| $(\frac{x}{(\cos(\Box))^2} + \sqrt{e^x})$ | $(((x)^2 \cdot \sin(\Box)) + e^{(x)^2})$ | $(x \cdot (\sqrt{x} - \sin((x + x))))$ |
| $\frac{\sqrt{\Box}}{(\sin((x - \Box)) - x)}$ | $(\frac{\frac{x}{\sqrt{x}}}{x} - e^\Box)$ | $(x + (x \cdot (\sqrt{\Box} + e^x)))$ |
| $(\frac{\Box}{(\Box - \sin(x))} + (e^x)^2)$ | $\frac{\sqrt{(\Box - (\sin(x) + \cos(\Box)))}}{x}$ | $((\Box - (x + \sqrt{(x)^2})) \cdot \cos(x))$ |
| $((\sqrt{x} - \Box) \cdot (\frac{\sqrt{x}}{\Box})^2)$ | $\frac{(\Box - ((x \cdot \sin(x)))^2)}{e^x}$ | $(\Box \cdot (\Box - (\sqrt{(\cos(x) - x)})^2))$ |
| $\frac{\Box}{(x \cdot \sqrt{(x - (\Box \cdot x))})}$ | $(\Box \cdot (x - (\sqrt{x} + \sqrt{e^x})))$ | $\frac{\sqrt{((\frac{x}{\Box})^2 + e^x)}}{\Box}$ |
| $(\sqrt{\frac{((x+x))^2}{\cos(x)}} - \Box)$ | $\frac{(e^{(\Box)^2} - \sin(x))}{(\sqrt{x})^2}$ | $(((\Box \cdot x))^2 + (e^{\frac{\Box}{x}})^2)$ |
| $\frac{(x - \sqrt{(x + e^\Box)})}{\sin(x)}$ | $\frac{(\cos(x) + \cos((x + x)))}{e^\Box}$ | $(((x + x) \cdot (\sin(\Box) - \Box)) - x)$ |

Table 8: Heldout expressions for $n_{var} = 1$. Part 2.

| | | |
|---|---|---|
| $(\cos(\square) - \sqrt{x})$ | $(\frac{\cos(x)}{x})^2$ | $(\square - \sqrt{e^x})$ |
| $((\square)^2 \cdot e^x)$ | $\frac{\sin(x)}{\cos(x)}$ | $(\cos(x) + e^\square)$ |
| $((x)^2 \cdot \sin(\square))$ | $(\frac{x}{\square} - x)$ | $(\square + \frac{\square}{x})$ |
| $(x \cdot \sqrt{\cos(x)})$ | $\sqrt{(\square \cdot \cos(x))}$ | $(\square \cdot \sqrt{(x)^2})$ |
| $\sqrt{(x \cdot e^\square)}$ | $\sqrt{(x - \cos(\square))}$ | $((\square - \cos(x)))^2$ |
| $(\square + \frac{\cos(x)}{\square})$ | $((\sin(\square) - \cos(x)))^2$ | $((\cos(x))^2 + \sin(x))$ |
| $\sqrt{(\square \cdot (x + x))}$ | $\frac{(e^x)^2}{e^\square}$ | $\frac{\sin(\square)}{(\square - x)}$ |
| $((\square \cdot x) - \sin(x))$ | $(x \cdot (\cos(x) - x))$ | $\frac{(\square - \sin(x))}{x}$ |
| $(x + (x - \sqrt{x}))$ | $((x - \square) \cdot e^\square)$ | $\frac{(\square + x)}{\cos(x)}$ |
| $\frac{(\square - \sqrt{x})}{x}$ | $\frac{\square}{(x + e^x)}$ | $(\frac{\square}{x} - (\square)^2)$ |
| $(\square + \sqrt{(x - \cos(\square))})$ | $(\frac{x}{\sqrt{\square}} - \cos(x))$ | $(\sqrt{(\square - x)} \cdot \sin(x))$ |
| $\frac{(\square)^2}{\sqrt{(\square - x)}}$ | $\frac{(\cos(\square) - \sin(\square))}{x}$ | $(\frac{(x - \cos(\square))}{x})^2$ |
| $\frac{(e^x - \sin(x))}{x}$ | $\frac{\sqrt{\square}}{(x + (x)^2)}$ | $(\square - \sqrt{\frac{\cos(\square)}{x}})$ |
| $\frac{\cos(\square)}{\sqrt{(x - \square)}}$ | $(x + (x - \frac{x}{\square}))$ | $(\square + \sqrt{\frac{(x)^2}{\square}})$ |
| $\frac{\sqrt{e^{(x-\square)}}}{x}$ | $(\sin((\square + \square)) + e^x)$ | $\frac{(\sin(x) - (\square)^2)}{x}$ |
| $((\square - x) \cdot \sqrt{(x + x)})$ | $(\sqrt{e^\square} - \frac{e^x}{\square})$ | $(x - ((\cos(x))^2 + \sin(x)))$ |
| $(x \cdot (x + \sin((x + x))))$ | $\frac{\frac{x}{\sqrt{e^\square}}}{e^x}$ | $((\sin(\square) \cdot e^x) - e^\square)$ |
| $\frac{(\square - \sin(\frac{x}{\square}))}{x}$ | $\frac{\sqrt{(e^x)^2}}{(\sin(\square))^2}$ | $((e^\square)^2 - \sin((\square + x)))$ |
| $(x \cdot (\square + \sqrt{(e^x)^2}))$ | $\frac{\square}{(\square \cdot ((x)^2 - x))}$ | $\frac{\frac{e^{(\frac{\square}{x})^2}}{\sqrt{x}}}{\frac{e^{(\square+x)}}{(\sin(\square))^2}}$ |
| $\frac{\square}{(\square + (x \cdot \sin(x)))}$ | $((\sin(x) \cdot e^x) - \sin(\square))$ | |
| $(\square - \sqrt{((\cos(\square))^2 + \cos(x))})$ | $\frac{(\frac{x}{\sin(x)} - (x)^2)}{\square}$ | $((\sqrt{(x + \sin(x))})^2 + \sqrt{\square})$ |
| $\sqrt{\frac{\frac{(\sqrt{\square} - \sqrt{\sin(\square)})}{e^x}}{((e^\square)^2 - \cos(x))}}$ | $\sqrt{((\cos(x))^2 - \frac{x}{e^\square})}$ | $\frac{x}{(\sqrt{\square} \cdot \cos((\square + x)))}$ |
| | $\frac{((\square \cdot x))^2}{\cos(\frac{\square}{x})}$ | $\frac{e^\square}{(\square + \cos((x + x)))}$ |
| $((e^x - \sin(\square)) - e^{(\square)^2})$ | $\sqrt{(((e^\square)^2 - x) \cdot (x)^2)}$ | $\frac{(\cos(x) + e^{(\square)^2})}{(x)^2}$ |
| $(x \cdot \frac{((x)^2 + \sqrt{x})}{\square})$ | $(x + \sqrt{\frac{x}{(\sin(\square) - \square)}})$ | $(\square - (x + (x \cdot \sqrt{\sin(x)})))$ |
| $\frac{(x - (\sqrt{x} \cdot e^x))}{\cos(\square)}$ | $\frac{x}{((\sqrt{\square} - \cos(x)) \cdot \cos(x))}$ | $(\sqrt{((\cos(x))^2 - x)} - (\sqrt{x})^2)$ |
| $(\frac{x}{\sin(\square)} - \sqrt{\frac{e^x}{\square}})$ | $((\square \cdot \frac{e^\square}{\sin(x)}) + e^x)$ | $\frac{(\square + \sqrt{(x - \sin(x))})}{\cos(x)}$ |
| $((\square + \frac{\square}{\sqrt{\cos((x+x))}}))^2$ | $(x - ((\square + (\sqrt{x})^2) \cdot \cos(\square)))$ | $(\frac{\frac{\sqrt{x}}{x}}{} + \sqrt{(x)^2})$ |
| $(\sqrt{(\square - e^x)} - \frac{\sin(x)}{x})$ | $(\sqrt{e^{\frac{(\square - x)}{x}}} - (x)^2)$ | $((x + (x)^2) \cdot \frac{\sin(x)}{\cos(x)})$ |
| $(((\square)^2 \cdot e^x) + \sin((\square \cdot x)))$ | $(\sqrt{((\sin(x) - \cos(\square)))^2} - (\square)^2)$ | $(\frac{(\frac{\sqrt{x}}{\square})^2}{x} - e^x)$ |

Table 9: Heldout expressions for $n_{var} = 2$. Part 1.

| | | |
|---|---|---|
| $(\cos(y) - e^x)$ | $\sqrt{\cos(\frac{y}{x})}$ | $\frac{\sqrt{e^y}}{x}$ |
| $((y)^2 - \sin(x))$ | $(\square + \frac{x}{y})$ | $(e^x - \sqrt{y})$ |
| $\sqrt{e^{\frac{x}{y}}}$ | $(\sqrt{\cos(x)} - y)$ | $(y - (\square \cdot x))$ |
| $((\square \cdot x) - y)$ | $(y - \sqrt{\cos(x)})$ | $(y + \sqrt{\cos(x)})$ |
| $(\sqrt{\sin(x)} - y)$ | $\frac{y}{(\sin(x))^2}$ | $\frac{(x-y)}{y}$ |
| $(x \cdot \sqrt{(y+y)})$ | $(y - (x \cdot \sin(y)))$ | $\frac{(x-y)}{\sqrt{\square}}$ |
| $\frac{x}{((\square - y))^2}$ | $(\sqrt{x} + \sqrt{\sin(y)})$ | $(\sqrt{y} + \sqrt{e^x})$ |
| $(y \cdot \frac{\cos(x)}{x})$ | $(x - \frac{y}{\cos(\square)})$ | $\frac{\sqrt{x}}{(\sin(y))^2}$ |
| $(y + \sqrt{(x+x)})$ | $\cos(\frac{(\square + y)}{x})$ | $((x - y) \cdot \cos(\square))$ |
| $(\square - (y + \cos(x)))$ | $(\frac{y}{x} - e^y)$ | $\frac{x}{(\sin(x) - y)}$ |
| $(\sin(x) - (y \cdot \sqrt{x}))$ | $\sqrt{(x + \cos(\frac{y}{\square}))}$ | $(\square - \frac{\sin(y)}{\sqrt{x}})$ |
| $(y + (x \cdot \sqrt{\sin(\square)}))$ | $(x - \frac{\sin(x)}{\sqrt{y}})$ | $(\sqrt{(y)^2} \cdot \sqrt{e^x})$ |
| $(\frac{\sin(x)}{\sin(y)} - x)$ | $((\square + (y \cdot \sin(x))))^2$ | $\frac{x}{(\square + \frac{y}{x})}$ |
| $\frac{((y)^2 - x)}{e^x}$ | $(y - \frac{\sin(x)}{\sin(\square)})$ | $\frac{e^{\frac{x}{(y)^2}}}{x}$ |
| $((y \cdot (\square + x)) - y)$ | $\frac{y}{(\sin(x) + \sin(x))}$ | $((x)^2 + \sin((y - \square)))$ |
| $(\sqrt{e^{(x)^2}} - (\sqrt{y})^2)$ | $((y + (\sqrt{\square} - e^x)))^2$ | $(x \cdot (x - (\frac{y}{\square})^2))$ |
| $((\frac{\sqrt{x}}{x} - \sqrt{y}))^2$ | $\frac{\sqrt{y}}{((x)^2 + \cos(y))}$ | $(e^y - \frac{(\square + x)}{\square})$ |
| $\frac{(\sin(\square) - \square)}{(x+y)}$ | $(((\square \cdot \sqrt{y}) + \sqrt{x}))^2$ | $(\sqrt{(y-x)} - \frac{\square}{x})$ |
| $(\cos((y \cdot (\square + x))) - \square)$ | $\frac{\sqrt{((\square)^2 - \sin(y))}}{x}$ | $(\square + (x + \sin((\square + y))))$ |
| $((y + \sqrt{(x)^2}) \cdot e^{\square})$ | $\frac{x}{((\square \cdot y) - \sin(\square))}$ | $\frac{(x - \frac{\cos(x)}{y})}{x}$ |
| $\frac{\sqrt{(\sin(y) + \cos(\square))}}{(x)^2}$ | $(x \cdot (\square - (\sin((\square + y)))^2))$ | $\frac{\sqrt{(y - \cos(x))}}{\sqrt{\sin(y)}}$ |
| $\frac{\frac{\sin((\square \cdot x))}{\sqrt{y}}}{y}$ | $(\sqrt{(\square \cdot y)} - \sqrt{\frac{x}{\square}})$ | $(\frac{(\square + \sqrt{\square})}{y} + e^x)$ |
| $\sqrt{\frac{(\sin(\square) + \cos(y))}{\sin(x)}}$ | $\frac{(\frac{y}{\sin(x)} + \cos(x))}{\square}$ | $\frac{((x)^2 + \sin(y))}{(\sqrt{x})^2}$ |
| $(\square \cdot (\frac{(x + \sin(\square))}{y})^2)$ | $((\square \cdot \sqrt{\frac{\square}{x}}) + \sin(y))$ | $(\frac{x}{\sqrt{\square}} - \frac{\sin(x)}{y})$ |
| $((y \cdot (y - \square)) - e^{(x)^2})$ | $\sqrt{((x+y) \cdot \frac{\sin(x)}{x})}$ | $\frac{\square}{\sqrt{\frac{\cos(y)}{(x+y)}}}$ |
| $\frac{(x - \frac{e^{\frac{\square}{y}}}{x})}{\frac{x}{x}}$ | $(x - (y \cdot \cos((\square + (y - x)))))$ | $(((y - e^y) \cdot \sin(x)) + e^{\square})$ |
| $\frac{e^{(\sqrt{y} - \square)}}{(y)^2}$ | $\frac{y}{((x \cdot (\sin(\square))^2) - \cos(x))}$ | $(\sqrt{\frac{(\cos(y))^2}{x}} + \sqrt{(y)^2})$ |
| $\frac{\frac{\sqrt{y}}{\square}}{((y)^2 + \cos(x))}$ | $(x + \frac{\cos(y)}{(\sqrt{x} + \sin(x))})$ | $((\square + \square) \cdot (\frac{y}{\sin(y)} - x))$ |
| $\frac{y}{((\sin(\square) - (x)^2) - \sin(y))}$ | $\frac{x}{((\square)^2 - \sqrt{(y + \cos(y))})}$ | $(x + (x + \sqrt{\frac{y}{(\sin(y))^2}}))$ |
| $\sqrt{(\frac{(x+y)}{\sin(\square)} + (\square)^2)}$ | $\frac{\sqrt{x}}{(x - (\sqrt{x} + \sin(y)))}$ | $(\square + (y \cdot (\sqrt{\sin(\frac{\square}{x})})^2))$ |

1404
1405
1406
1407
1408
1409
1410

Table 10: Heldout expressions for $n_{var} = 2$. Part 2.

1411
1412
1413
1414
1415
1416
1417
1418
1419
1420
1421
1422
1423
1424
1425
1426
1427
1428
1429
1430
1431
1432
1433
1434
1435
1436
1437
1438
1439
1440
1441
1442
1443
1444
1445
1446
1447
1448
1449
1450
1451
1452
1453
1454
1455
1456
1457

| | | |
|---|---|---|
| $(y - \frac{x}{y})$ | $\sqrt{\frac{\cos(y)}{x}}$ | $((x)^2 \cdot \sin(y))$ |
| $(\sqrt{x} \cdot \sqrt{y})$ | $\sqrt{\frac{\sin(x)}{y}}$ | $\sqrt{\frac{e^x}{y}}$ |
| $\frac{y}{\sqrt{\sin(x)}}$ | $e^{\frac{y}{(x)^2}}$ | $((e^x)^2 - y)$ |
| $\frac{e^x}{\sin(y)}$ | $(\sqrt{x} - (y)^2)$ | $(y \cdot (x - \square))$ |
| $\frac{\cos(y)}{\cos(x)}$ | $(x + \sqrt{(y)^2})$ | $\frac{x}{(\square + y)}$ |
| $\frac{y}{\sqrt{(\square + x)}}$ | $\frac{(\square - x)}{\sin(y)}$ | $(x \cdot \sin(\frac{x}{y}))$ |
| $\frac{(y - \cos(x))}{\square}$ | $((x \cdot e^\square) - y)$ | $\frac{\cos((\square + x))}{y}$ |
| $\frac{x}{(\sin(y) - y)}$ | $\sqrt{((y \cdot \sin(x)))^2}$ | $\frac{x}{(y + \sin(\square))}$ |
| $\cos((x - \frac{y}{\square}))$ | $\frac{(\cos(y) - y)}{x}$ | $(\sin(y) - \sqrt{\sin(x)})$ |
| $(x \cdot \cos(\frac{\square}{y}))$ | $(((x + y))^2 - x)$ | $(x + \cos(\frac{\square}{y}))$ |
| $\sqrt{(y - \cos((\square + x)))}$ | $((y + (\sin(x) - \square)))^2$ | $(y \cdot \frac{(x)^2}{\sin(x)})$ |
| $(x - ((\square \cdot \sqrt{y}))^2)$ | $\frac{\sqrt{(\square + y)}}{(x)^2}$ | $(\sqrt{\square} - \frac{(y)^2}{x})$ |
| $\sqrt{\frac{x}{(y + \cos(y))}}$ | $\frac{(x + e^x)}{(y)^2}$ | $(\sqrt{x} - \frac{y}{\sqrt{\square}})$ |
| $\sqrt{\cos((x \cdot (y + y)))}$ | $(y + \sqrt{\frac{\cos(x)}{\square}})$ | $((\square + e^{(x - y)}))^2$ |
| $\frac{((x)^2 - (y)^2)}{\square}$ | $(\frac{x}{e^\square} + (y)^2)$ | $((\square)^2 - \frac{y}{(x)^2})$ |
| $\frac{y}{\sqrt{(\sin(y) \cdot e^x)}}$ | $\frac{(\frac{y}{\square})^2}{x}$ | $\frac{\frac{\cos(x)}{y}}{\sqrt{e^y}}$ |
| $((x \cdot y) + \cos(\frac{\square}{\square}))$ | $(\square \cdot ((x)^2 \cdot \sqrt{(y)^2}))$ | $\frac{(\square + e^x)}{(x \cdot y)}$ |
| $(\frac{\sqrt{e^y}}{x} - (y)^2)$ | $(\frac{e^{(y - x)}}{\sin(\square)})^2$ | $(x + \frac{\frac{y}{\sqrt{x}}}{y})$ |
| $(\frac{(x)^2}{\sin(x)} - \sin(y))$ | $(y + (\frac{y}{x} + (\square)^2))$ | $\frac{\sin(x)}{\sqrt{(x - \sin(y))}}$ |
| $\frac{((\sqrt{y} - \cos(x)))^2}{x}$ | $((\cos(x))^2 - (x \cdot \sqrt{y}))$ | $(y - (y \cdot \frac{\sin(x)}{x}))$ |
| $(\sqrt{\frac{x}{(e^x)^2}} - \sqrt{y})$ | $((x - \sqrt{\frac{x}{(\square + y)}}))^2$ | $\frac{(\cos(y) - \square)}{\sin((\square + x))}$ |
| $((x \cdot y) - (\sqrt{x} \cdot \sin(y)))$ | $((\sqrt{(y)^2} - (\square)^2) - \cos(x))$ | $(((\square + \square) \cdot e^{\frac{x}{y}}))^2$ |
| $(\sqrt{\frac{\square}{e^y}} + \sqrt{e^x})$ | $\frac{(x - \sqrt{y})}{\cos((\square + x))}$ | $((y \cdot (x - y)) - (\sin(\square))^2)$ |
| $(\frac{e^x}{\square} - (y \cdot \sqrt{y}))$ | $(\square \cdot (e^{(\square - y)} - e^x))$ | $((\sqrt{x} + \sqrt{\sin(y)}) + \sin(\square))$ |
| $(((\sin(x) - y) - y) - e^\square)$ | $((y + (\frac{\cos(y)}{x} - x)))^2$ | $\sqrt{\frac{(\cos(y) - (y)^2)}{\sin(x)}}$ |
| $\frac{x}{(y + (\sin((\square \cdot x)) - \square))}$ | $\frac{\sqrt{x}}{(y - \sqrt{\frac{x}{e^\square}})}$ | $(\frac{\square}{(x)^2} + \sqrt{(y - \cos(x))})$ |
| $(x + (x + ((x \cdot y) - e^\square)))$ | $(\sin((y - x)) - \frac{\cos(x)}{\sqrt{y}})$ | $((\sqrt{\frac{e^x}{y}} - x) - \cos(y))$ |
| $((\square \cdot \frac{(x)^2}{\sqrt{y}}) \cdot \cos(\square))$ | $\frac{y}{((y - (\sin(\square))^2) \cdot (x)^2)}$ | $(y + \frac{\square}{\sin(\frac{x}{(\square \cdot y)})})$ |
| $(x + ((\frac{\square}{\sin(y)})^2 \cdot \sin(\square)))$ | $((x - (e^y + e^y)) \cdot \sin(x))$ | $\frac{((x \cdot \sqrt{\sin(y)}) - e^x)}{y}$ |
| $((\sin((x \cdot y)) \cdot e^\square) - \cos(x))$ | $(((\square - \frac{x}{\sin(y)}))^2 + \cos(y))$ | $((\frac{\cos(y)}{\square})^2 + \sqrt{(\square \cdot x)})$ |

Table 11: Heldout expressions for $n_{var} = 3$. Part 1.

| | | |
|---|---|---|
| $(y + (z - x))$ | $((x \cdot z) - y)$ | $\frac{y}{(x+z)}$ |
| $(x + \frac{y}{z})$ | $\frac{(y-z)}{x}$ | $(\frac{z}{y} - x)$ |
| $(x + (y \cdot z))$ | $(y \cdot (x - z))$ | $(y \cdot \frac{z}{x})$ |
| $(z - (x \cdot y))$ | $\frac{(y+z)}{x}$ | $\frac{(x-z)}{y}$ |
| $\frac{x}{(y+z)}$ | $\frac{z}{(x-y)}$ | $\frac{z}{(x \cdot y)}$ |
| $e^{\frac{(y+z)}{x}}$ | $\frac{x}{(\sqrt{y}-z)}$ | $(z \cdot \sin(\frac{y}{x}))$ |
| $(z - \sin((x+y)))$ | $((\frac{x}{y})^2 - z)$ | $((y \cdot \sin(x)) - z)$ |
| $(y \cdot e^{(x-z)})$ | $\frac{\sin((x \cdot z))}{y}$ | $\frac{z}{(y \cdot \cos(x))}$ |
| $\frac{y}{e^{\frac{x}{z}}}$ | $((y \cdot e^x) - z)$ | $(\frac{(x+y)}{z})^2$ |
| $\sin((x + \frac{z}{y}))$ | $(x - (y + \sin(z)))$ | $((y + z) \cdot \cos(x))$ |
| $\frac{(\sin(z)-y)}{\sqrt{x}}$ | $\frac{z}{(x+(x-y))}$ | $(\sin(\frac{y}{x}) - \sin(z))$ |
| $(\sqrt{x} \cdot \sqrt{(z - y)})$ | $(x \cdot \frac{(x+y)}{z})$ | $(\sqrt{(z + e^x)} - y)$ |
| $\frac{(y-\sqrt{\cos(x)})}{z}$ | $((z - y) - e^{(x)^2})$ | $\frac{(y+e^x)}{\sqrt{z}}$ |
| $(x + (z + \sqrt{\cos(y)}))$ | $\frac{(\sin(y))^2}{(z-x)}$ | $\frac{(x+z)}{(x \cdot y)}$ |
| $((y \cdot \frac{\sqrt{x}}{z}))^2$ | $(\frac{\sqrt{y}}{z} - \cos(x))$ | $\frac{x}{(y \cdot (x-z))}$ |
| $\frac{z}{\cos((y \cdot (x-y)))}$ | $(x - (z + \cos((y + y))))$ | $\sqrt{(\frac{x}{(y)^2} - \cos(z))}$ |
| $\frac{e^{(z \cdot (x-z))}}{y}$ | $((z)^2 + ((x - \cos(y)))^2)$ | $(e^x - \frac{x}{(y \cdot z)})$ |
| $\frac{z}{(\sin(\frac{x}{y})-x)}$ | $\sqrt{\frac{(z-(x+y))}{\Box}}$ | $(z + ((\Box + x) \cdot (y)^2))$ |
| $(\sqrt{y} - (\cos((x + z)))^2)$ | $\frac{x}{(y \cdot \sin((y+z)))}$ | $(\sqrt{x} + \sqrt{((y - z))^2})$ |
| $\frac{z}{(\cos((x+z))-y)}$ | $\frac{x}{((z \cdot \sin(y))-z)}$ | $\frac{(\Box+x)}{\cos((y-z))}$ |
| $(\frac{z}{(\cos(\frac{x}{y}))^2} - z)$ | $(x \cdot \frac{e^{(z-y)}}{(y)^2})$ | $(\sqrt{(x - z)} - (x \cdot (y)^2))$ |
| $\frac{((\sqrt{y} \cdot \sin(z))-x)}{z}$ | $((z \cdot (y + \sin(\frac{x}{\Box}))))^2$ | $((z \cdot \cos((\Box - x))) + \sin(y))$ |
| $((x \cdot \frac{z}{\sqrt{y}}) + e^z)$ | $(\frac{y}{\sin(z)} + \sqrt{(x - y)})$ | $(\Box \cdot (\sqrt{(\cos(y) - x)} - z))$ |
| $(x + (e^{\frac{x}{z}} - \cos(y)))$ | $(y - (\sqrt{(x - z)} \cdot e^{\Box}))$ | $\frac{\Box}{(y \cdot e^{((x-z))^2})}$ |
| $\frac{(y-z)}{(\sqrt{x}-(z)^2)}$ | $(x \cdot (\cos(z) - \frac{y}{\cos(\Box)}))$ | $(x - \frac{\Box}{(z+(\sin(y))^2)})$ |
| $((x + (y \cdot \sqrt{\sin(\Box)})) - e^z)$ | $(e^{(x-\Box)} - \sqrt{(y + \cos(z))})$ | $(((e^x - \Box) + e^y) \cdot \sqrt{z})$ |
| $\frac{\sin(z)}{(\cos(y)-\frac{z}{\sin(x)})}$ | $(((x)^2 - \sin(z)) + \sin((\Box + y)))$ | $\frac{z}{((e^{(z-y)})^2+\sin(x))}$ |
| $\frac{\sqrt{\sin(x)}}{(x+\sqrt{(z-y)})}$ | $((x + \frac{(\Box)^2}{z}) \cdot (x - y))$ | $\frac{e^{\frac{y}{z}}}{\sqrt{(y+(x)^2)}}$ |
| $(x + (\Box \cdot \frac{\sin(\frac{z}{y})}{y}))$ | $(y - \frac{\sin((x \cdot z))}{e^{(\Box)^2}})$ | $\frac{(\sqrt{y}+e^{(\Box \cdot z)})}{\sqrt{x}}$ |
| $\sqrt{\frac{((y-\frac{z}{\sin(x)}))^2}{z}}$ | $((((y)^2 - z) + \cos(x)) \cdot \sqrt{x})$ | $(y \cdot \frac{z}{\cos((\Box+(x+z)))})$ |

Table 12: Heldout expressions for $n_{var} = 3$. Part 2.

| | | |
|---|---|---|
| $(\frac{x}{y} - z)$ | $(y + \frac{z}{x})$ | $(x \cdot (z - y))$ |
| $(x \cdot (y - z))$ | $(y - (x \cdot z))$ | $(x \cdot \frac{y}{z})$ |
| $(x - (y + z))$ | $(x \cdot (y \cdot z))$ | $(z - \frac{x}{y})$ |
| $(y - \frac{z}{x})$ | $\frac{x}{(z-y)}$ | $(x - (y \cdot z))$ |
| $\frac{x}{(y-z)}$ | $(y - \frac{x}{z})$ | $(z - (x + y))$ |
| $(x + (z - \sin(y)))$ | $(\frac{(y)^2}{z} - x)$ | $((x - \frac{z}{y}))^2$ |
| $(\frac{z}{x} - \sqrt{y})$ | $((x \cdot z) + e^y)$ | $\frac{(z-x)}{(y)^2}$ |
| $e^{(x - \frac{z}{y})}$ | $\frac{\sin((y-x))}{z}$ | $((x \cdot \sqrt{z}) - y)$ |
| $\frac{x}{(z+\sqrt{y})}$ | $((z)^2 - \frac{x}{y})$ | $((z - y) \cdot (x)^2)$ |
| $((y \cdot z) + \sin(x))$ | $(y \cdot \frac{\sin(x)}{z})$ | $\frac{z}{(y-\cos(x))}$ |
| $(x + \frac{z}{(\Box - y)})$ | $\frac{\sqrt{(x+e^z)}}{y}$ | $(x - (z + \frac{y}{x}))$ |
| $(x - (\sin(y) + \cos(z)))$ | $(e^z \cdot e^{\frac{x}{y}})$ | $\frac{(z+z)}{(x \cdot y)}$ |
| $(x - (\sqrt{(y - z)})^2)$ | $\frac{(z+\frac{y}{x})}{\Box}$ | $\frac{(x+\sin(z))}{\sin(y)}$ |
| $(\frac{(y-x)}{x} - z)$ | $((y \cdot \sqrt{(z - x)}))^2$ | $\frac{y}{e^{\frac{z}{(x)^2}}}$ |
| $(e^{(y-x)} - e^z)$ | $\frac{\sin(z)}{\sqrt{(y-x)}}$ | $(((x)^2 - y) \cdot (z)^2)$ |
| $(\frac{(\sin(y))^2}{\sqrt{z}} - x)$ | $(z \cdot (\frac{x}{(x-y)})^2)$ | $((y + \frac{\cos(z)}{\cos(x)}))^2$ |
| $((z)^2 + \sqrt{\frac{x}{\cos(y)}})$ | $\frac{(\cos((\Box \cdot y)) - x)}{z}$ | $(\Box \cdot \frac{(x-y)}{e^z})$ |
| $(\sqrt{\cos(y)} - \sqrt{(x \cdot z)})$ | $\sqrt{((x + \sin(y)) \cdot \sin(z))}$ | $(x + (\sqrt{(\Box + y)} - z))$ |
| $\sqrt{(\frac{(x)^2}{y} + \sin(z))}$ | $((y \cdot z) - \frac{\cos(y)}{x})$ | $(\Box - \frac{z}{(x+(y)^2)})$ |
| $((z \cdot (z + \sin(x))) - y)$ | $(((\Box + z) \cdot e^x) - y)$ | $\frac{(\Box + \frac{\cos(z)}{x})}{y}$ |
| $\frac{(x)^2}{(\frac{y}{z} + \sin(y))}$ | $(x - \frac{\frac{\sqrt{e^x}}{y}}{z})$ | $\sqrt{\cos(\frac{x}{(y \cdot (x+z))})}$ |
| $(((\cos(x))^2 - \sin(y)) \cdot (z)^2)$ | $\sqrt{\frac{((x)^2 - \frac{x}{y})}{z}}$ | $\frac{y}{(x \cdot \sqrt{\frac{\cos(\Box)}{z}})}$ |
| $((x \cdot ((y)^2 - y)) + (z)^2)$ | $(\frac{\sqrt{(x \cdot \sin(y))}}{x} - z)$ | $\frac{(\Box + x)}{\sqrt{\frac{e^z}{y}}}$ |
| $(y \cdot ((x + z) \cdot (\sin(y))^2))$ | $\sqrt{(z - (x \cdot ((\Box + y))^2))}$ | $(z - (y \cdot (z + (\cos(x))^2)))$ |
| $(((\frac{y}{z})^2 \cdot \cos(y)) - x)$ | $(\frac{(y - \cos(z))}{(x-z)})^2$ | $((x - (y \cdot \cos(\frac{z}{x}))))^2$ |
| $(z \cdot (\sqrt{x} - (y \cdot (y + z))))$ | $(y - \frac{\frac{(z)^2}{\sqrt{x}}}{(z)^2})$ | $(e^{\frac{x}{z}} - (y \cdot \sqrt{\sin(\Box)}))$ |
| $(x + \frac{z}{((\sqrt{y} + \sin(z)))^2})$ | $\frac{(y - (\Box + (z \cdot \sin(y))))}{x}$ | $(\frac{\sqrt{\sin(x)}}{y} + (\frac{z}{y})^2)$ |
| $(\Box \cdot (x + (x \cdot \frac{\cos(y)}{z})))$ | $\frac{x}{((\sin(x))^2 + \sqrt{\frac{z}{y}})}$ | $(y + \frac{\sin(z)}{(x - \frac{y}{x})})$ |
| $(\sqrt{(x + \frac{(x-z)}{x})} - y)$ | $(y - (z \cdot (\Box + (x \cdot \sqrt{y}))))$ | $(\sqrt{z} - (x \cdot \sqrt{\frac{x}{e^y}}))$ |
| $\frac{x}{((z)^2 \cdot \sqrt{((\Box + y))^2})}$ | $(z + (\frac{(y-x)}{\Box} + \sin(y)))$ | $(\sqrt{\cos(x)} + \sin((y \cdot (x - z))))$ |

Table 13: Heldout expressions for $n_{var} = 4$. Part 1.

| | | |
|---|---|---|
| $\frac{z}{(w-\frac{y}{x})}$ | $((w \cdot (x+y)) - z)$ | $\frac{x}{((y \cdot w)-z)}$ |
| $(\frac{x}{w} - \frac{z}{y})$ | $((z \cdot \frac{w}{y}) - x)$ | $\frac{x}{(w-\frac{y}{z})}$ |
| $\frac{(x-\frac{w}{z})}{y}$ | $\frac{z}{((y \cdot w)-x)}$ | $(x \cdot \frac{y}{(z-w)})$ |
| $(\frac{w}{(y+z)} - x)$ | $(z \cdot (x - \frac{y}{w}))$ | $(\frac{w}{(x \cdot z)} - y)$ |
| $\frac{w}{(x+(y \cdot z))}$ | $\frac{(y+z)}{(x-w)}$ | $(y \cdot (z - (x+w)))$ |
| $(\frac{y}{z} - (w \cdot e^x))$ | $((w \cdot \cos(z)) + \frac{x}{y})$ | $(e^{(w \cdot (x-z))} - y)$ |
| $(\frac{\frac{\cos(z)}{x}}{y} - w)$ | $(z + \frac{(w+\sin(y))}{x})$ | $\sqrt{(x \cdot (z + \frac{y}{w}))}$ |
| $e^{(x \cdot \frac{(y-w)}{z})}$ | $\frac{y}{(x-(w+\sin(z)))}$ | $((w \cdot ((y)^2 - z)) - x)$ |
| $(x + \frac{(z+\sin(w))}{y})$ | $(y \cdot ((z-w) \cdot \sqrt{x}))$ | $(y \cdot \frac{(z)^2}{(w-x)})$ |
| $(w \cdot (x + \cos((y-z))))$ | $((z \cdot ((x-w))^2) - y)$ | $(z + e^{\frac{x}{(w-y)}})$ |
| $(w - ((\frac{z}{x} - \cos(y)))^2)$ | $\frac{(\sin(\frac{z}{w})-x)}{e^y}$ | $((x \cdot \sqrt{e^z}) + (y \cdot w))$ |
| $\frac{\sqrt{(x \cdot (y+(w)^2))}}{z}$ | $(\frac{y}{(w)^2} + \frac{\sqrt{x}}{z})$ | $(x \cdot \sqrt{(y \cdot \frac{\sin(w)}{z})})$ |
| $(\frac{(y-x)}{\sqrt{z}} - \sqrt{w})$ | $((\cos(z) - y) - ((x-w))^2)$ | $\frac{(\frac{w}{\cos(y)}-x)}{\cos(z)}$ |
| $(y \cdot (z + e^{((x-w))^2}))$ | $\frac{(w+\sin(y))}{(x-\cos(z))}$ | $(((y-w) \cdot (\cos(x))^2) - z)$ |
| $(w - ((x - \sqrt{(y+z)}))^2)$ | $\sqrt{\frac{((z+\frac{w}{y}))^2}{x}}$ | $((\sqrt{w} - x) \cdot ((y+z))^2)$ |
| $\frac{(\frac{(y-\square)}{w}+\cos(x))}{z}$ | $((y \cdot (x+w)) + e^{(z-w)})$ | $(x - e^{\frac{(y-z)}{(w-z)}})$ |
| $\frac{\frac{y}{e^z}}{e^{(\frac{x}{w})^2}}$ | $(\frac{(\frac{z}{\cos(y)})^2}{x} - (w)^2)$ | $(z + e^{\frac{w}{(\frac{x}{y}-y)}})$ |
| $(\square \cdot \frac{(z+\cos(x))}{(y-w)})$ | $\sqrt{\frac{((y)^2+((z \cdot w))^2)}{x}}$ | $(y + (\sqrt{(x \cdot w)} - (\square \cdot z)))$ |
| $(y + ((z \cdot w) - \frac{\sin(x)}{z}))$ | $(\frac{\sqrt{w}}{z} - \frac{y}{\sqrt{(x)^2}})$ | $((z - (x \cdot (\sqrt{w})^2)) \cdot \cos(y))$ |
| $\frac{(\sin(\frac{y}{(x-z)})-w)}{z}$ | $((\sqrt{z} - (y)^2) \cdot ((x \cdot w))^2)$ | $((x+z) \cdot \frac{e^{(y-w)}}{z})$ |

Table 14: Heldout expressions for $n_{var} = 4$. Part 2.

| | | |
|---|---|---|
| $(w \cdot (y + \frac{z}{x}))$ | $((y \cdot (z - x)) - w)$ | $((y \cdot z) - \frac{x}{w})$ |
| $(y - \frac{(x+w)}{z})$ | $(\frac{y}{(x \cdot w)} - z)$ | $(x - (y \cdot \frac{z}{w}))$ |
| $\frac{(x-y)}{(z \cdot w)}$ | $\frac{(\frac{w}{y}-z)}{x}$ | $(y \cdot (\frac{x}{z} - w))$ |
| $(\frac{w}{(x+z)} - y)$ | $(x \cdot (\frac{y}{w} - z))$ | $(w - (x + (y \cdot z)))$ |
| $\frac{(z+(x \cdot y))}{w}$ | $\frac{w}{(\frac{z}{x}-y)}$ | $(w \cdot (x - \frac{y}{z}))$ |
| $\sin(\frac{(x-z)}{(y+w)})$ | $\frac{z}{(y \cdot (w \cdot (x)^2))}$ | $(w \cdot (y + (\sin(x) - z)))$ |
| $\frac{(w+\frac{z}{\sin(x)})}{y}$ | $((w \cdot (e^y - x)) - z)$ | $(e^{\frac{x}{(y \cdot z)}} - w)$ |
| $(y \cdot (z + e^{(x-w)}))$ | $\sqrt{(x + (y \cdot (w - z)))}$ | $\frac{(x+(\cos(y)-w))}{z}$ |
| $\sqrt{\frac{y}{(w-(x \cdot z))}}$ | $\frac{(z+\sqrt{w})}{(x \cdot y)}$ | $\frac{(x+((y \cdot w))^2)}{z}$ |
| $\frac{(y-w)}{(x-(z)^2)}$ | $(((y \cdot z) - x) - \cos(w))$ | $(y + \frac{x}{(\sqrt{w}-z)})$ |
| $(x + (w + \frac{(\cos(y))^2}{z}))$ | $(\frac{x}{(y-(z \cdot w))} - y)$ | $\frac{((\sqrt{y}-z)-(w)^2)}{x}$ |
| $(\frac{(w-\sin(x))}{(y)^2} - z)$ | $\frac{\sqrt{y}}{(w+\sin(\frac{x}{z}))}$ | $((w \cdot (y - \frac{x}{z})) - y)$ |
| $(\frac{(\frac{x}{w})^2}{z} \cdot \sin(y))$ | $(w \cdot (\frac{x}{y} - \frac{\square}{z}))$ | $\frac{x}{(\frac{y}{\sqrt{w}}+\sin(z))}$ |
| $(x \cdot (((y)^2 \cdot \cos(z)) - w))$ | $(w + e^{\frac{y}{((x)^2-z)}})$ | $((y \cdot \cos(z)) - ((x \cdot w))^2)$ |
| $\frac{(w-\cos(x))}{(y+\cos(z))}$ | $\frac{z}{((\sin(w)-x)+\cos(y))}$ | $((x + (\cos(y) - \frac{w}{z})))^2$ |
| $((((x - \cos(y)))^2 \cdot \sin(z)) - w)$ | $((\square - z) \cdot \cos(\frac{x}{(y+w)}))$ | $(z - \frac{\cos((\square+(x+y)))}{w})$ |
| $((z)^2 \cdot \sqrt{(w + \frac{(x)^2}{y})})$ | $((\frac{\sqrt{y}}{(w)^2} - z) + \sqrt{x})$ | $(\frac{(\sin((y+w)))^2}{z} + \sin(x))$ |
| $(\sqrt{(y + ((z \cdot w) - x))} - z)$ | $\frac{w}{(\square - \frac{z}{\cos(\frac{x}{y})})}$ | $\frac{x}{(\frac{e^z}{\sin(w)}-\cos(y))}$ |
| $\frac{\sqrt{(\frac{e^w}{y}-\sin(z))}}{x}$ | $(((y \cdot \frac{\sin(w)}{x}))^2 \cdot \sqrt{z})$ | $(z - \frac{\sqrt{\frac{y}{(z-x)}}}{w})$ |
| $((y \cdot (\frac{y}{x})^2) - \frac{z}{w})$ | $((\square - (z \cdot \sqrt{x})) \cdot (y - w))$ | $((\frac{w}{y} + \sqrt{(x + e^z)}))^2$ |