# OpenReview forum: "Symbolic Regression with Self-Supervised Heuristic Beam Search"
_ICLR.cc/2026/Conference — ICLR 2026 Conference Withdrawn Submission_

### Official Review · Reviewer_WgPo · 2025-10-30

**Soundness:** 3
**Presentation:** 3
**Contribution:** 3
**Rating:** 4
**Confidence:** 3

**Summary:**

This paper presents HTSSR, a novel symbolic regression method combining beam search with a self-supervised heuristic. The approach is interesting, and the focus on interpretability and parsimony is commendable. The scalability analysis across several dimensions is a valuable contribution. However, I have several major concerns regarding the evaluation methodology, the scope of the experiments, and the claims made, which I detail below.

**Strengths:**

The core idea of using a self-supervised learned heuristic to guide a beam search is both novel and elegant. It represents a clear departure from common paradigms, such as using fitness-to-data as a direct reward signal in reinforcement learning or generating expressions token-by-token with a generative model. The concept of learning a "precedence" relation between expressions is a powerful way to inject learned knowledge into the search process.

**Weaknesses:**

**1. Major Concerns​**

​**1.1. Evaluation Metric: Over-reliance on Symbolic Solution Rate (SSR)​​**

The authors argue against using R² for evaluation, favoring the Symbolic Solution Rate (SSR). While SSR is appropriate for problems with known ground truth, it has significant limitations for real-world, black-box problems where the true expression is unknown. Furthermore, SSR is an extremely stringent metric. It may unfairly penalize expressions that are functionally equivalent or very close to the true solution, which could be easily corrected with minor adjustments based on domain knowledge by human or some other mechanisms. A more holistic evaluation philosophy should prioritize achieving high fitting accuracy first, and then, under that constraint, minimize expression complexity and maximize out-of-distribution generalization. Relying solely on SSR presents an idealized and potentially unrealistic assessment of practical utility.

​**1.2. Limited Noise Robustness Evaluation​**

The tested noise levels (0.0 and 0.01) are too low to adequately demonstrate the method's robustness. Pre-trained models can be sensitive to noise; small noise levels may have a negligible impact, while performance can degrade significantly as noise increases. The results in Figure 6 suggest that, apart from AI Feynman, HTSSR's performance drops noticeably even with the minimal 0.01 noise level, indicating potential vulnerability to data imperfections. A more convincing demonstration of robustness would require testing against higher noise levels.

​**1.3. Scalability Analysis is Incomplete​**

The scalability analysis, while a strength of the paper, misses critical dimensions:

​Expression Length:​​ The tested maximum length of 10 symbols is insufficient. With higher input dimensions (e.g., 6 variables), even a simple additive model can approach this length. The method's performance on longer, more complex expressions remains unverified.

​Variable Dimensionality:​​ The search space grows combinatorially with the number of variables. The paper does not sufficiently investigate how performance scales as the number of variables increases beyond 3 or 4. This is a crucial aspect of scalability, as each additional variable significantly expands the search space and likely demands more training data.

**​1.4. Claims of Sample Efficiency are Unsubstantiated​**

The introduction claims the method is "sample-efficient," but no experimental evidence is provided to support this claim. The experiments in Section 4.1 analyze the effect of the number of evaluationdata points on search success, not the efficiency of the trainingprocess. Furthermore, the appendix indicates non-trivial time开销, which seems to contradict the notion of high sampling efficiency. An experiment demonstrating effective learning with limited training data or a comparison of training data requirements against other methods is needed.

​**2. Methodological Concerns​**
​**2.1. Risk of Overfitting in Pre-training​**

HTSSR relies on pre-training a heuristic model on synthetic data, effectively teaching it "correct" expression paths. This approach carries the risk of poor out-of-distribution generalization. If the synthetic data distribution does not align well with the target problems, the pre-trained heuristic may misguide the search, leading to significant performance drops on unseen equation types or domains.

**Questions:**

The questions are described in section "Weaknesses"

---

> ### Author Response · Authors · 2025-11-28
> **Rebuttal**
>
> We thank the reviewer for the comments.
>
> **General Comment.** We now have a much more complete set of experiments on SRBench and fixed ablation experiments.
> - Blackbox and ground-truth problems
> - Reporting with the traditional SRBench plots (Pareto, R2, model size, running time, etc.)
> - All of the random states trials
> - Noise levels 0.0, 0.01, and 0.1
> - More works added to the direct comparison, like SR4MDL [1], E2ESR [2], and NeuralSR [3].
> - Ablations fixed to isolate the contribution of the changing parts.
> - In the blackbox datasets, our method figures in the Pareto frontier of R2 x model size.
> - In the ground-truth datasets, our method is indeed competitive, surpassing GP-based methods and some learning-based too.
>
> **About the weaknesses of the paper**
>
> **1. Major Concerns**
>
> **​1.1. Evaluation Metric: Over-reliance on Symbolic Solution Rate (SSR)​​** We accept the recommendation of using $R^2$, as we run our method on the blackbox datasets of SRBench and report $R^2$ along with other useful metrics like model size, running time.
>
> **​1.2. Limited Noise Robustness Evaluation​** Our new set of experiments now includes the higher $0.1$ noise level from SRBench.  Our new results do show robustness to noise level up to $0.01$. At noise level $0.1$ there is noticeable degradation, but still better than most of the GP-based and some learning-based methods at noise $0.0$.
>
> **​1.3. Scalability Analysis is Incomplete​** As of now, we understand that the length reach of HTSSR is a combination of both search strategy and heuristic model quality. We have a pre-set experiment where a simulated heuristc derived from flipping the true precedence signal with some probability gives some notion about the reachability at each level of accuracy (or recall of classes 0 and 1). We suspect that if this flipping noise happens at about 15% for both positive and negative classes, the expressions of length 10 and beyond start to look dificult in the beam search. In this pre-set experiment we also have a stochastic search implemented to compare against the beam search under the same simulated heuristic. Results indicate that no search algorithm dominates the other in the broader sense. MCTS would be interesting to add to the comparison.
> If permitted, we commit to add this heuristic simulation experiment to the text. That would include larger expressions and more variables.
>
> **1.4. Claims of Sample Efficiency are Unsubstantiated​** We commit to rephrase "sample efficient" and analogous claims to something closer to what we really mean, which is the ability of the heuristic to learn with just a few observed data points, but not necessarily being sample efficient in the training sense.
>
> **2. Methodological Concerns​​**
>
> **2.1. Risk of Overfitting in Pre-training​** Because the heuristic model is meant to be dedicated to the given observed dataset and also because the expression formation has almost no assumption about data distribution, except for the constant placeholder, we believe that overfitting can onlly happen if we train on expressions up to a size $n$ and try to use the heuristic to find expressions of bigger size not seen during training. We do think about having a generalist model, but we use a dedicated version in this work.

---

### Official Review · Reviewer_PE48 · 2025-10-31

**Soundness:** 3
**Presentation:** 3
**Contribution:** 3
**Rating:** 4
**Confidence:** 4

**Summary:**

The paper proposes HTSSR (HeurisTic beam Search for Symbolic Regression): a self-supervised learned heuristic classifier that guides beam search through the expression space to find short, interpretable formulas from data, including formulas with free constants. It evaluates on SRBench (Feynman/Strogatz) and reports competitive performance, plus an analysis of scalability with respect to expression length, number of variables, number of domains, and number of samples.

**Strengths:**

Clear, modular design. Expressions are generated by a prefix grammar; the learned heuristic performs a simple precedence (pairwise ordering) classification instead of modeling discrete structures directly.
Solid engineering. The paper introduces Sort-Diff and Digit numeric feature transforms to stabilize training under non-differentiability and scale/units issues.
Efficient constant fitting. Free constants are optimized with a small Levenberg–Marquardt inner loop, typically converging in few steps.
Stricter evaluation. It emphasizes symbolic solution rate (SSR) over loose R^2 thresholds and discusses shortcomings of prior evaluation practices.
Sample efficiency & robustness. Can recover ground-truth formulas with as few as ~10 samples and shows better robustness to moderate noise on Strogatz tasks.
Comprehensive scalability study. Systematically examines how success rate changes with length, variables, domains, and samples, with useful practical takeaways.

**Weaknesses:**

Attribution of gains is unclear. The paper notes that improvements from Sort-Diff may partly stem from increased parameter count, making ablations hard to interpret.
Strong dependence on the primitive set. Results and “difficulty” trends are assessed under a fixed set of primitives; changing this set may alter conclusions, limiting external validity.
Length scalability bottleneck. Success rate drops sharply for expressions longer than ~10 under fixed budgets, constraining practical problem complexity.
Unequal trial counts vs. baselines. HTSSR is run once per task whereas baselines often run 10×; confidence intervals are not directly comparable and may bias rankings.
Sample-efficiency test is narrow. The detailed sample-size curve is shown on a single domain; broader validation would strengthen the claim.
Multi-domain trade-offs. Training across more domains can degrade target performance in some settings; reuse across domains is not conclusively demonstrated.
Stopping/acceptance tolerance sensitivity. The relative-error tolerance (e.g., 10^{-3}) is fixed to accommodate noise; its effect on rankings is not fully analyzed.
Robustness tweaks may bias numerics. Safety operators and clipping for stability could distort certain physical relationships; more sensitivity analysis would help.

**Questions:**

See weakness

**Details Of Ethics Concerns:**

NO or VERY MINOR ethics concerns only

---

> ### Author Response · Authors · 2025-11-27
> **Rebuttal**
>
> We thank the reviewer for the comments.
>
> **General Comment.** We now have a much more complete set of experiments on SRBench and fixed ablation experiments.
> - Blackbox and ground-truth problems
> - Reporting with the traditional SRBench plots (Pareto, R2, model size, running time, etc.)
> - All of the random states trials
> - Noise levels 0.0, 0.01, and 0.1
> - More works added to the direct comparison, like SR4MDL [1], E2ESR [2], and NeuralSR [3].
> - Ablations fixed to isolate the contribution of the changing parts.
> - In the blackbox datasets, our method figures in the Pareto frontier of R2 x model size.
> - In the ground-truth datasets, our method is indeed competitive, surpassing GP-based methods and some learning-based too.
>
> **About the weaknesses of the paper**
>
> **Attribution of gains is unclear. The paper notes that improvements from Sort-Diff may partly stem from increased parameter count, making ablations hard to interpret.** We have fixed the ablation experiments to make sure the neural nets have the same size specification. Ablations are now easy to interpret and changes are isolated.
>
> **Strong dependence on the primitive set. Results and “difficulty” trends are assessed under a fixed set of primitives; changing this set may alter conclusions, limiting external validity** In the paper, when we talk about the use of a single primitive set, we mean that this is a problem of Symbolic Regression in the general sense, not particularly of our method. We commit to rephrase this better in the paper. The vocubulary/primitives from which the expressions are formed is a decision to be made before the search process and indeed might result in very different results. Regarding our choice of primitives, it is like any other commonly used, containing basic operations like +, -, *, /, sin, cos, exp, and sqrt.
>
> **Success rate drops sharply for expressions longer than ~10 under fixed budgets, constraining practical problem complexity.** As of now, we understand that the length reachability of HTSSR is a combination of both search strategy and heuristic model quality. We have a pre-set experiment where a simulated heuristc derived from flipping the true precedence signal with some probability gives some notion about the reachability at each level of accuracy (or recall of classes 0 and 1). We suspect that if this flipping noise happens at about 15% for both positive and negative classes, the expressions of length 10 and beyond start to look dificult in the beam search. In this pre-set experiment we also have a stochastic search implemented to compare against the beam search under the same simulated heuristic. Results indicate that no search algorithm dominates the other in the broader sense. MCTS would be interesting to add to the comparison.
>
> **Unequal trial counts vs. baselines. HTSSR is run once per task whereas baselines often run 10×; confidence intervals are not directly comparable and may bias rankings.** We have fixed this issue in the updated results.
>
> **Sample-efficiency test is narrow. The detailed sample-size curve is shown on a single domain; broader validation would strengthen the claim.** We agree with the reviewer, but we must say that the variability of domains is, at least indirectly, tested in the new results on SRBench, as there are many different datasets in blackbox and ground-truth.
>
> **Stopping/acceptance tolerance sensitivity. The relative-error tolerance (e.g., 10^{-3}) is fixed to accommodate noise;** We have slightly changed the acceptance criteria to have a new, smaller, fixed tolerance $2 \cdot 10^{-4}$ and let the search track the best so far for the case it ends without meeting the tolerance.

---

### Official Review · Reviewer_CkoB · 2025-11-04

**Soundness:** 2
**Presentation:** 2
**Contribution:** 1
**Rating:** 2
**Confidence:** 5

**Summary:**

This paper proposes Self-Supervised Heuristic Beam Search (SS-HBS) to improve the efficiency and structural generalization of Symbolic Regression (SR).
The method learns a heuristic function that estimates the probability of a partial expression being structurally valid.
This heuristic is trained in a self-supervised manner and used to guide the beam search process, aiming to focus exploration on promising symbolic structures.

Empirical results on SRBench show improvements over prior SR methods.

**Strengths:**

- The idea of integrating structural knowledge into the beam search process is interesting and could make SR exploration more efficient.
- The self-supervised heuristic formulation is conceptually appealing when explicit supervision on expression quality is unavailable.
- The proposed method achieves improvements on noisy data.

**Weaknesses:**

1. Lack of theoretical and experimental comparison with generative Transformer models such as E2ESR.

   In models like E2ESR, the generative probability can directly serve as a heuristic for search.
   The proposed heuristic seems conceptually similar to this posterior, yet the authors do not justify why another heuristic function is necessary.

2. Insufficient comparative analysis.

    No comparison with alternative self-supervised learning strategies (e.g., synthetic-data-based pretraining).
    Ablation studies are limited, making it difficult to isolate which component contributes most to the improvement.

3. Weak theoretical grounding.

   It is unclear whether the proposed method is meaningful or empirically justified.
   Sections 3 is too abstract; the motivation for the loss function and input representation is not well explained.
   The appendices contain only minimal implementation details, making full reproducibility difficult.

**Questions:**

I would like the authors to address the points listed under Weaknesses.

---

> ### Author Response · Authors · 2025-11-27
> **Rebuttal, Part 1**
>
> We thank the reviewer for the comments.
>
> **General Comment.** We now have a much more complete set of experiments on SRBench and fixed ablation experiments.
>
> - Blackbox and ground-truth problems
> - Reporting with the traditional SRBench plots (Pareto, R2, model size, running time, etc.)
> - All of the random states trials
> - Noise levels 0.0, 0.01, and 0.1
> - More works added to the direct comparison, like SR4MDL [1], E2ESR [2], and NeuralSR [3].
> - Ablations fixed to isolate the contribution of the changing parts.
> - In the blackbox datasets, our method figures in the Pareto frontier of R2 x model size.
> - In the ground-truth datasets, our method is indeed competitive, surpassing GP-based methods and some learning-based too.
>
> **About the weaknesses of the paper**
>
> **1. Lack of theoretical and experimental comparison with generative Transformer models such as E2ESR.** As pointed out in the general comment, we improved our experimental comparisons, including generative Transformer-based models like E2ESR. We also commit ourselves to add theoretical comparison with such class of models.
>
> **1. [...] In models like E2ESR, the generative probability can directly serve as a heuristic for search. The proposed heuristic seems conceptually similar to this posterior, yet the authors do not justify why another heuristic function is necessary.** One key difference is that the generative probability from models like E2ESR are with respect to tokens, thus inducing a search for expressions in the space of sequences of tokens. Our method imposes an ordering in the set of expressions such that each new element is itself a complete expression and has an associated probability of precedence to the target. In principle, this reduces the number of inferences necessary to attribute a probability to a search element and dedicates the model to this task, not making the expression symbols directly.
>
> Our heuristic model can be combined, very easily, with any search strategy like the simple - yet highly parallelizable - stochastic search, beam search, MCTS, and so on. Yet, most recent token-based learn + search methods seem to put a lot of emphasis on MCTS specifically.
>
> **2. Insufficient comparative analysis. No comparison with alternative self-supervised learning strategies (e.g., synthetic-data-based pretraining).** With the updated results we now have direct comparisons with other self-supervised learning strategies, specially SR4MDL [1], E2ESR [2], and Neural SR [3]. Regarding theoretical comparison, we commit ourselves to include such methods in the text.
>
> **2. [...] Ablation studies are limited, making it difficult to isolate which component contributes most to the improvement.** We fixed the ablation experiments with issues about attribution of gains, specifically the Sort-Diff (Appendix A2, Figure 7) and the Sample Efficiency (Section4.1, Figure 3). We did that by padding the shorter inputs to the maximum used lenght, making all ablation experiments use the same neural net size specification.
>
> Now we observe the SSR for Sort-Diff:
>
> | Size | No Sort-Diff (Old $\rightarrow$ New) | Sort-Diff (kept) |
> |---|---|---|
> | $5$ |	$1.00 \rightarrow 1.00$	 | $1.00$ |
> | $6$ |	$0.97 \rightarrow 0.97$	 | $1.00$ |
> | $7$ |	$0.81 \rightarrow 0.78$    | $0.89$ |
> | $8$ |	$0.64 \rightarrow 0.58$    | $0.66$ |
> | $9$ |	$0.19 \rightarrow 0.26$    | $0.43$ |
> | $10$ |	$0.07 \rightarrow 0.21$    | $0.23$ |
>
> And for sample efficiency (Old $\rightarrow$ New):
>
> | Size | $10^1$ | $10^2$ | $10^3$ (kept) |
> |---|---|---|---|
> | $5$ |	$1.00 \rightarrow 1.00$ | $1.00 \rightarrow 1.00$	 | $1.00$ |
> | $6$ |	$1.00 \rightarrow 0.97$ | $0.97 \rightarrow 1.00$	 | $0.97$ |
> | $7$ |	$0.85 \rightarrow 0.92$ | $0.93 \rightarrow 0.85$	 | $0.85$ |
> | $8$ |	$0.39 \rightarrow 0.50$ | $0.69 \rightarrow 0.61$	 | $0.66$ |
> | $9$ |	$0.19 \rightarrow 0.24$ | $0.36 \rightarrow 0.37$	 | $0.41$ |
> | $10$ |	$0.00 \rightarrow 0.04$ | $0.15 \rightarrow 0.19$	 | $0.20$ |
>
> We also add a new ablation, where we compare using our Digit Transform versus an adapted version of the sign-mantissa-exponent representation commonly used and introduced in [2]:
>
> | Size |	Ours |	Kamienny et al., 2022 |
> |---|---|---|
> | 5	| 1.00	| 1.00 |
> | 6	| 0.97	| 0.94 |
> | 7	| 0.78	| 0.73 |
> | 8	| 0.58	| 0.55 |
> | 9	| 0.26	| 0.15 |
> | 10	| 0.21	| 0.04 |
>
> **We address Weakness 3 in the next comment**
>
> [1] Symbolic Regression Via Mdlformer-guided Search: From Minimizing Prediction Error To Minimizing Description Length.
>
> [2] End-to-end Symbolic Regression with Transformers.
>
> [3] Neural Symbolic Regression that Scales.

---

> ### Author Response · Authors · 2025-11-27
> **Rebuttal, Part 2**
>
> **About the weaknesses of the paper**
>
> **[...]**
>
> **3. Weak theoretical grounding. It is unclear whether the proposed method is meaningful or empirically justified** Now with the extended results we have more empirical justification for the method, as we see competitive performance in SRBench, for both blackbox and ground-truth problem sets. Furthermore, we think that HTSSR brings more than scores in the benchmark:
>
> - A simple and modular design, allowing easy customization and evaluation: the heuristic can be trained indepedently from the search algorithm, besides being easy to integrate. Also, the primitives, grammar rules, and formation constraints are easy to specify and can be adapted freely. In contrast, parametric models that do generate symbols are prone to generating malformed expressions.
>
> - Direct, synthesizeable, and dense objective to be learned by the heuristic: some recent works that use pre-training, like SR4MDL [1] and DSO [5] need multiple training phases and the design of complicated rewards [5]. HTSSR only needs BCE loss and a single training phase.
>
> - Fast and broad constant fitting: we prove with the extensive experiments that the inner optimization with Levenberg-Marquardt is fast and capable of finding constants in non-trivial cases, as in the held-out dataset. Other methods generally relly on BFGS [2] (or some variant) or assume a limited form of constant instantiation [1]. We believe that for a small number of variables (< 10), Levenberg-Marquardt converges with much fewer iterations than BFGS.
>
> - Adoption of a dedicated over general-purpose heuristic that can learn the particularities of a problem and does not need to assume any form of data distribution (excepting when evaluating the constant placeholder).
>
> **3. [...]  Sections 3 is too abstract; the motivation for the loss function and input representation is not well explained.** The choice of loss function is motivated by the fact that the search problem is framed in a way that puts the expressions in a order-like relation, which in turn can be seen as a set of pairs. The binary classification objective then tries to predict if a give pair belongs to the relation. Regarding the input representation, some recent works use the token-based representation from [2], which uses 4 significant digits and can represent in the order of $10^8$ values, which might be insufficient for the vast space of expressions. HTSSR uses its Digit Transform motivated by training stability (values in a controlled range) and representation power (7 significant places for float 32 bit and 16 for 64 bit float). Also, our comparison provided in Part 1 showed dominance of our representation over that from [2], at least in our held-out dataset.
>
> **On a more technical aspect of theoretical justification**
>
> If the heuristic prediction is the true relation, a beam search that prioritizes expressions based on precedence value and size will, at every step of the search, contain at least one expression that precedes the target. Also, each new search round will produce slightly larger expressions, until a solutions is found. The parsimoneous increase in size of preceding expressions then guarantees that a solution with minimum size will be found.
>
> In principle, search methods that have lookahead, like MCTS or a simple stochastic search, can go through rollout paths that find solutions with non-minimum size. Therefore, beam search would find more concise expressions by design. In this context, HTSSR does decrease the edit distance to target expression monotonically and thus has an optimal substructure like SR4MDL [1].
>
> The theoretical justification for HTSSR follows a similar argument to that in SR4MDL [1] and reffering to (Cormen et al., 2022) [7]: the search objective also has an optimal substructure.
>
> Finally, MDL is proven to be incomputable in the general sense (Vitanyi, 2020) [8], framing the heuristic as a precedence classification model is a way to try to avoid such hardship, at least in principle.
>
> [2] End-to-end Symbolic Regression with Transformers.
>
> [5] Deep Symbolic Optimization: Reinforcement Learning for Symbolic Mathematics
>
> [7] Introduction to Algorithms, Fourth Edition
>
> [8] How incomputable is kolmogorov complexity?

---

### Official Review · Reviewer_jk6z · 2025-11-09

**Soundness:** 3
**Presentation:** 1
**Contribution:** 1
**Rating:** 2
**Confidence:** 5

**Summary:**

In this paper, the authors propose a sample-efficient symbolic regression method, named HTSSR, which aims to recover exact expressions from observed data points. The approach begins with a source expression and progressively expands it into more complex expressions by applying a set of predefined rules. To identify the most promising expansion path, the authors apply a beam search algorithm guided by a heuristic model trained to estimate the likelihood that an expression can be expanded to the ground-truth expression. The authors analyze the scalability of the proposed approach and evaluate its performance in recovering exact expressions on two white-bow problem sets, Feynman and Strogatz.

**Strengths:**

The authors propose a simple yet effective symbolic regression method that combines beam search with a heuristic, achieving sample efficiency while yielding interpretable and concise expressions. Experiments on two white-box problem sets from SRBench demonstrate the method’s ability to discover exact expressions from a small set of observation sample points. Further experiments also demonstrate the scalability of the proposed approach in terms of expression length, number of variables, number of domains, and number of observed data points.

**Weaknesses:**

**W1: Insufficient literature review and weak novelty.** The proposed method is highly similar to SR4MDL[1], a work published at ICLR 2025, which also starts from a source expression, expands it into more complex formulas by applying a series of predefined rules, and employs a heuristic-guided Monte Carlo tree search to identify an optimal expansion path. In SR4MDL, the heuristic is provided by a pre-trained Transformer that learns to predict the minimal number of transformations required to convert a given expression into a target one, which seems to be comparable to the heuristic model proposed by the author. Furthermore, SR4MDL is also evaluated on the Feynman and Strogatz problem sets to assess its symbolic solution rate. Given the strong methodological and experimental overlap, the absence of any discussion or performance comparison with SR4MDL in the present work appears unacceptable and concerning.

**W2: Insufficient experiments and limited applicability.** Although the authors claim that their method is designed for discovering exact expressions, in most symbolic regression scenarios, the underlying expression is either unknown or may not even exist. Therefore, in addition to evaluations on white-box datasets, the proposed method should also be tested on black-box datasets to assess its ability to discover expressions balancing accuracy and complexity. Without demonstrating its ability to identify concise and accurate formulas from data with unknown underlying relationships, its practical applicability cannot be determined.

**W3.** Given that the proposed method involves not only a search procedure but also the training of a heuristic model, although the authors provide the runtime in the appendix, the time required to discover the target expressions should be compared with other SR methods and discussed in more detail in the main text.

[1] Symbolic regression via MDLformer-guided search: from minimizing prediction error to minimizing description length (ICLR, 2025)

**Questions:**

Q1. Although the authors conduct a series of ablation studies, it seems that no experiments ablate the heuristic model itself from the beam search. Could the authors provide an ablation experiment to demonstrate the contribution of the heuristic model to the performance?

Q2. In Line 250, instead of adopting the digit transform approach proposed in (Kamienny et al., NeurIPS 2022) and widely adopted by many SR works, the authors design their own digit transform approach. Could the authors provide a comparison of their performances?

Q3. In Figure 2, what does $V^\*(E)$ represent? I understand that $E$ denotes the expression to be evaluated, but it seems that the definition of $V^\*$ is not provided in the text.

Q4. In the input to the Heuristic Model, how are the numerical constants in $E$ determined? If the constants are directly fitted on $\mathcal{D}$ using the method described in Section 3.4, wouldn’t that cause expressions that can, in fact, expand to F to be incorrectly classified as 'cannot'? For example, if $F = 2x + 1$ and $E = □ \times x$, numerical fitting may yield $E = 2.5x$, leading the model to judge that $E$ cannot be expanded into $F$ since $2.5 ≠ 2$, even though such an expansion is theoretically possible.

---

> ### Author Response · Authors · 2025-11-27
> **Rebuttal, Part 1**
>
> We thank the reviewer for the review and for pointing out the SR4MDL paper [1]. We, the authors, found it after the submission deadline. Answers are listed below.
>
> **General Comment.** We now have a much more complete set of experiments on SRBench and **fixed** ablation experiments.
> - Blackbox and ground-truth problems
> - Reporting with the traditional SRBench plots (Pareto, R2, model size, running time, etc.)
> - All of the random states trials
> - Noise levels 0.0, 0.01, and 0.1
> - More works added to the direct comparison, like SR4MDL, E2ESR [2], and NeuralSR [3].
> - Ablations fixed to isolate the contribution of the changing parts.
> - In the blackbox datasets, our method figures in the Pareto frontier of R2 x model size.
> - In the ground-truth datasets, our method is indeed competitive, surpassing GP-based methods and some learning-based too.
>
> **Q1. Although the authors conduct a series of ablation studies, it seems that no experiments ablate the heuristic model itself from the beam search. Could the authors provide an ablation experiment to demonstrate the contribution of the heuristic model to the performance?** We believe that the experiment in Figure 3 does that to some extent, as the $10^3$ dummy line is in fact a heuristic that was not trained, just initialized weights. The comparison against models trained using $10$, $10^2$, and $10^3$ data points from the observed data shows that using the trained heuristic models increases SSR.
>
> **Q2. In Line 250, instead of adopting the digit transform approach proposed in (Kamienny et al., NeurIPS 2022) and widely adopted by many SR works, the authors design their own digit transform approach. Could the authors provide a comparison of their performances?** We performed an experiment comparing our digit transform with an equivalent of the representation in (Kamienny et al., NeurIPS 2022). Consider that their representation is token-based, while ours is not, but we adapted their sign-mantissa-exponent to our neural network. Results on the held-out dataset show dominance in perormence of our representation, being:
>
> | Size | Ours | Kamienny et al., 2022 |
> |---|---|---|
> | 5  | 1.00  | 1.00  |
> | 6  | 0.97  | 0.94  |
> | 7  | 0.78  | 0.73  |
> | 8  | 0.58  | 0.55  |
> | 9  | 0.26  | 0.15  |
> | 10  | 0.21  | 0.04  |
>
> **Q3. In Figure 2, what does V\*(E) represent? I understand that E denotes the expression to be evaluated, but it seems that the definition of V\* is not provided in the text.** V\*(E) represents the evaluation of the expression tree of E at each node/subtree, such that the values in the root are the values of the complete expression. For instance, E\*("/ x y") contains E("x"), E("y"), and E("/ x y"). We commit to add clarification to the paper.
>
> **Q4. In the input to the Heuristic Model, how are the numerical constants in E determined?** The expression evaluations that feed the heuristic model can: (i) attribute a default value for ◻ (e.g. 1) **or** (ii) attribute a random value from a pre-defined random distribution (e.g. U(-1, 1)).
>
> **Q4. [...] If the constants are directly fitted on D using the method described in Section 3.4, wouldn’t that cause expressions that can, in fact, expand to F to be incorrectly classified as 'cannot'? For example, if F = 2\*x +1 and E = ◻\*x, numerical fitting may yield E = 2.5\*x, leading the model to judge that E cannot be expanded into F since $2.5 \neq 2$, even though such an expansion is theoretically possible.** The fitting described in Section 3.4 only happens after the expression is evaluated by the heuristic. Also, the expansion is top-down (leaves are expanded), not bottom-up like in [1], therefore, in the example you provided, E = ◻\*x cannot expand into F = 2\*x +1 not because constant fitting would mislead the search, but because E and F are structurally different from a top-down perspective.
>
> **On the weaknesses of the paper**
>
> **W1: Insufficient literature review and weak novelty.** We commit to add more papers that also adopt the learning strategy plus search or one-shot prediction, like [1], [2], [3], SymFormer [4], DSO [5], and TPSR [6]. Regarding novelty, we explain in the next comment some key differences from SR4MDL and other closely-related works that HTSSR brings.
>
> **W2: Insufficient experiments and limited applicability** As shown in the general comment above, we have substantially improved our experiments in many aspects, including trials, metrics, and more datasets.
>
> **W3.** We now have the runtimes and can promptly add to the main text, directly comparing with other methods.
>
> [1] Symbolic Regression Via Mdlformer-guided Search: From Minimizing Prediction Error To Minimizing Description Length.
>
> [2] End-to-end Symbolic Regression with Transformers.
>
> [3] Neural Symbolic Regression that Scales.
>
> [4] SymFormer: End-to-end symbolic regression using transformer-based architecture
>
> [5] Deep Symbolic Optimization: Reinforcement Learning for Symbolic Mathematics
>
> [6] Transformer-based Planning for Symbolic Regression

---

> ### Author Response · Authors · 2025-11-27
> **Rebuttal, Part 2**
>
> We commit to improve our related work and include SR4MDL in our comparisons.
>
> Regarding conceptual similarity with SR4MDL, both indeed learn a model in a self-supervised way that guides a search afterwards. Also, both include the structural aspect in the learning objective of the heuristic and avoid using fitness to data as a learning signal.
>
> When it comes to the conceptual differences and novelties that HTSSR brings, we note:
>
> - SR4MDL tries to predict the MDL of the target values with respect to current expressions in the search pool.
> New expressions are formed from previous ones, creating increasingly complex expressions in a bottom-up fashion. On the other hand, HTSSR predicts a formation precedence and forms new expressions by expanding leaf nodes of a given expression, a top-down approach.
> - SR4MDL seems to be trained as general purpose and makes assumptions about the distribution of data. HTSSR is thought to be dedicated to the data at hand and needs to be trained from zero for every new dataset. Despite the lack of reusability, the advatange is the dedication to the search problem at hand without making assumptions about the distribution of data, except for ◻.
> - SR4MDL models a numeric value in a two-phase training procedure, needing to align latent spaces. HTSSR models a simple binary category and can be trained in a single phase with BCE loss.
> - SR4MDL fits contants when the search is one-step away from the solution, but only for a pre-defined form (e.g. a linear combination where constants are the coeficients). HTSSR can fit constants in a broader class of expressions, like $e^{1.45 x}$ and $\sin (2.79 + x)$, and is limited only by the primitives, generation rules, and formation (complexity) constraints. On a side note, the Strogatz and Feynman datasets contain mostly the easier cases for constant fitting.
> - SR4MDL uses the token-based representation from [2], which uses 4 significant digits and can represent in the order of 10^8 values. HTSSR uses its Digit Transform motivated by training stability (values in a controlled range) and representation power (7 significant places for float 32 bit and 16 for 64 bit float).
>
>
> **About theoretical justification**
>
> If the heuristic prediction is the true relation, a beam search that prioritizes expressions based on precedence value and size will, at every step of the search, contain at least one expression that precedes the target. Also, each new search round will produce slightly larger expressions, until a solutions is found. The parsimoneous increase in size of preceding expressions then guarantees that a solution with minimum size will be found.
>
> In principle, search methods that have lookahead, like MCTS or a simple stochastic search, can go through rollout paths that find solutions with non-minimum size. Therefore, beam search would find more concise expressions by design.
> In this context, HTSSR does decrease the edit distance to target expression monotonically and thus has an optimal substructure too, but notice that MDL of target with respect to the current expressions does not make sense in HTSSR.
>
> The theorethical justification for HTSSR follows a similar argument to that in SR4MDL and reffering to (Cormen et al., 2022) [7]: the search objective also has an optimal substructure.
>
> Finally, MDL is proven to be incomputable in the general sense (Vitanyi, 2020) [8], framing the heuristic as a precedence classification model is a way to try to avoid such hardship, at least in principle.
>
> [7] Introduction to Algorithms, Fourth Edition
>
> [8] How incomputable is kolmogorov complexity?

---

### Author Response · Authors · 2025-11-28
**Information**

We would like to tell the reviewers that we are currently finishing to add the improvements to the text and will upload the new version soon.

---

### Author Response · Authors · 2025-12-03
**Information**

We are grateful for the suggestions and questions from the reviewers.

We have now updated the text after reflecting on the suggestions.

Best regards,

The authors.

---

### Note · Authors · 2026-02-10

I have read and agree with the venue's withdrawal policy on behalf of myself and my co-authors.

---

### Meta-Review · Area_Chair_oZiC · 2026-01-07

**Summary:**

While reviewers acknowledge solid engineering and an appealing learned-heuristic-guided search idea, they collectively raise major concerns about insufficient novelty relative to close prior work (notably SR4MDL), missing key comparisons (SR4MDL, generative Transformer SR), limited and potentially idealized evaluation (SSR-heavy, mostly white-box), inadequate ablations and reproducibility, and unclear efficiency/robustness/scalability claims.

**Reviewer Concerns:**

In their response, the authors present supplementary experiments on the suggested baselines and metrics. However, several key issues regarding novelty relative to SR4MDL and idealized evaluation remain.

**Reviewer Scores:**

I think the first two reviewers might increase their ratings to 4 in response to the authors' supplementary experiments. Overall, the average rating is still rejected.

---

### Decision · Program_Chairs · 2026-01-26

Reject